# The plant pathogen *Pseudomonas aeruginosa* triggers a DELLA-dependent seed germination arrest in *Arabidopsis*

Hicham Chahtane[1,2], Thanise Nogueira Füller[1,2], Pierre-Marie Allard[3], Laurence Marcourt[3], Emerson Ferreira Queiroz[3], Venkatasalam Shanmugabalaji[1,2†], Jacques Falquet[4], Jean-Luc Wolfender[3], Luis Lopez-Molina[1,2*]

[1]Department of Plant Biology, University of Geneva, Geneva, Switzerland; [2]Institute for Genetics and Genomics in Geneva, University of Geneva, Geneva, Switzerland; [3]School of Pharmaceutical Sciences, EPGL, University of Geneva, University of Lausanne, Geneva, Switzerland; [4]University of Geneva, Geneva, Switzerland

**Abstract** To anticipate potential seedling damage, plants block seed germination under unfavorable conditions. Previous studies investigated how seed germination is controlled in response to abiotic stresses through gibberellic and abscisic acid signaling. However, little is known about whether seeds respond to rhizosphere bacterial pathogens. We found that *Arabidopsis* seed germination is blocked in the vicinity of the plant pathogen *Pseudomonas aeruginosa*. We identified L-2-amino-4-methoxy-trans-3-butenoic acid (AMB), released by *P. aeruginosa*, as a biotic compound triggering germination arrest. We provide genetic evidence that in AMB-treated seeds DELLA factors promote the accumulation of the germination repressor ABI5 in a GA-independent manner. AMB production is controlled by the quorum sensing system IQS. In vitro experiments show that the AMB-dependent germination arrest protects seedlings from damage induced by AMB. We discuss the possibility that this could serve as a protective response to avoid severe seedling damage induced by AMB and exposure to a pathogen.
DOI: https://doi.org/10.7554/eLife.37082.001

*For correspondence:
luis.lopezmolina@unige.ch

Present address: †Laboratoire de Physiologie Végétale, Institute of Biology, Université de Neuchâtel, Neuchâtel, Switzerland

Competing interests: The authors declare that no competing interests exist.

## Introduction

Seeds are remarkable structures promoting plant dispersal by preserving the plant embryo in a desiccated and highly resistant state. Their appearance in the course of land plant evolution is regarded as a cornerstone of the striking spread and diversification of angiosperms among terrestrial plants.

Seed imbibition with water is the necessary first step to permit germination, transforming the embryo into a fragile juvenile seedling. However, upon imbibition, the seed is also exposed to potentially fatal environmental conditions for the future seedling. To avoid premature death, plants have evolved control mechanisms that block germination under unfavorable conditions to maintain the highly protected embryonic state (*Lopez-Molina et al., 2001*; *2002*).

Historically, studies focused on how seeds respond to abiotic stresses such as high temperature, canopy light -unfavorable for photosynthesis- or high salinity (*Baskin and Baskin, 1998*; *Reynolds and Thompson, 1971*; *Negbi et al., 1968*). In *Arabidopsis*, perception of abiotic factors leads to changes in seed endogenous levels of gibberellic acid (GA) and abscisic acid (ABA), two phytohormones playing a central role to control germination (*Cutler et al., 2010*; *Lau and Deng, 2010*; *Nonogaki, 2014*; *Davière and Achard, 2016*). Under favorable conditions, GA synthesis increases, which is necessary to initiate germination. GA induces proteolysis of DELLA factors repressing germination, which are encoded by a family of five genes: *RGL2, GAI, RGA, RGL1* and *RGL3*.

**eLife digest** The plant embryo within a seed is well protected. While it cannot stay within the seed forever, the embryo can often wait for the right conditions before it develops into a seedling and continues its life cycle. Indeed, plants have evolved several ways to time this process – which is known as germination – to maximize the chances that their seedlings will survive. For example, if the environment is too hot or too dark, the seed will make a hormone that stops it from germinating.

In addition to environmental factors like light and temperature, a seed in the real word is continuously confronted with soil microbes that may harm or benefit the plant. However, few researchers have asked whether seeds control their germination in response to other living organisms.

The bacterium *Pseudomonas aeruginosa* lives in a wide spectrum of environments, including the soil, and can cause diseases in both and plants and animals. Chahtane et al. now report that seeds of the model plant *Arabidopsis thaliana* do indeed repress their germination when this microbe is present. Specifically, the seeds respond to a molecule released from the bacteria called L-2-amino-4-methoxy-trans-3-butenoic acid, or AMB for short. Like the bacteria, AMB is harmful to young seedlings, but Chahtane et al. showed that the embryo within the seed is protected from its toxic effects.

Further experiments revealed that the seed's response to the bacterial molecule requires many of the same signaling components that repress germination when environmental conditions are unfavorable. However, Chahtane et al. note that AMB activates these components in an unusual way that they still do not understand.

The genes that control the production of AMB are known to also control how bacterial populations behave as they accumulate to high densities. It is therefore likely that *Pseudomonas aeruginosa* would make AMB if it reached a high density in the soil. This raises the possibility that plants have specifically evolved to stop germination if there are enough microbes nearby to pose a risk of disease. This hypothesis, however, is only one of several possible explanations and remains speculative at this stage; further work is now needed to evaluate it. Nevertheless, identifying how AMB interferes with the signaling components that control germination and plant growth may guide the design of new herbicides that could, for example, control weeds in the farming industry.
DOI: https://doi.org/10.7554/eLife.37082.002

DELLA factors collectively repress germination by promoting the accumulation of ABA and ABA-response transcription factors (TFs) ultimately repressing seed germination such as ABI3 and ABI5 (*Lee et al., 2002*; *Penfield et al., 2006*; *Lee et al., 2010a*; *Lopez-Molina and Chua, 2000*; *Lopez-Molina et al., 2001*; *2002*; *Finkelstein and Lynch, 2000*; *Piskurewicz et al., 2008*; *2009*). RGL2 can play a prominent role among DELLAs to stimulate ABA signaling and thus repress germination. Indeed, only *rgl2* mutants can germinate when seeds are treated with a GA synthesis inhibitor (*Lee et al., 2002*; *Piskurewicz et al., 2008*). This is likely due to the positive regulation of *RGL2* mRNA levels by ABA, which generates a positive feedback loop sustaining high RGL2 accumulation relative to other DELLAs (*Piskurewicz et al., 2008*, *2009*). How DELLA factors stimulate ABA signaling in seeds remains to be understood. DELLAs are unable to directly bind to DNA and thus are more likely to interact with TFs or other factors regulating ABA signaling. Recent work has shown that DELLA protein activity is regulated through phosphorylation, SUMOylation, O-GlcNAcylation, or O-fucosylation (*Conti et al., 2014*; *Qin et al., 2014*; *Zentella et al., 2016*, *2017*).

Along with abiotic factors, seeds are also continuously confronted to bacteria, fungi and animals (e.g. nematodes) present in soil and potentially acting as pathogens or commensals (*Silby et al., 2011*). Little is known about whether biotic factors released by non-plant organisms induce seed germination responses in plants. To address this question in the model plant *Arabidopsis*, we considered the case of *Pseudomonas*, a genus of Gram-negative bacteria having pathogenic and commensal interactions with both animals and plants (*Silby et al., 2011*).

We confronted *Arabidopsis* seeds to different *Pseudomonas* species and found a strong germination repressive activity (GRA) released by *Pseudomonas aeruginosa*. Strikingly, *P. aeruginosa* did not repress the germination of *Arabidopsis* mutant seeds lacking DELLA factors or ABA signaling

components. Metabolomic and bioguided biochemical fractionation approaches led us to identify the oxyvinylglycine L-2-amino-4-methoxy-trans-3-butenoic acid (also referred as methoxyvinylglycine or AMB) as the main GRA released by *P. aeruginosa*. AMB production and release is dependent on the five-gene operon *ambABCDE* controlling the newly identified quorum-sensing IQS in *P. aeruginosa* (*Lee et al., 2010b*, *2013a*). Using synthetic AMB, we provide genetic evidence that the activity of DELLAs to stimulate ABA-dependent responses is enhanced in seed exposed to AMB. Whether this reflects a mechanistic link between AMB and DELLAs is not known. Furthermore, AMB induces severe developmental defects in juvenile seedlings. In contrast, germination-arrested seeds are capable to produce viable plants when no longer exposed to AMB or *P. aeruginosa*.

Oxyvinylglycines are known to inhibit irreversibly pyridoxal phosphate (PLP)-dependent enzymes (*Berkowitz et al., 2006*). Furthermore, AMB is a methionine analog. Our results suggest that AMB does not block germination by interfering with ethylene, auxin or methionine synthesis. Rather, our results indicate that AMB interferes with an unknown and GA-independent mechanism promoting DELLA-dependent germination arrest, which involves DELLA-dependent stimulation of ABA signaling. We discuss possible interpretations of our findings, including the possibility that the germination arrest could serve as a protective response to avoid severe seedling damage induced by AMB and exposure to a pathogen.

## Results

### *Arabidopsis* seed germination is repressed by *Pseudomonas aeruginosa*

We explored whether *Pseudomonas* bacteria release compounds inhibiting *Arabidopsis* seed germination as follows: (1) individual *Pseudomonas* species were propagated for 3 days on germination agar medium supplemented with a carbon source; (2) thereafter *Arabidopsis* seeds were sown on the germination medium at various distances from the bacteria; (3) germination was scored after culturing seeds for 3 days (Materials and methods, *Figure 1—figure supplement 1A*). Germination was not markedly repressed by *P. fluorescens, P. putida*, *P. syringae* or *Escherichia coli* (used as a Gram-negative non-*Pseudomonas* species control) (*Figure 1A and B*). In marked contrast, germination was strongly repressed by *P. aeruginosa* (WT strains *PAO1* and *PA14*) (*Figure 1A and B*). The percentage of germination increased with increasing distance separating seeds and *P. aeruginosa* (*Figure 1A* and *Figure 1—figure supplement 1B*). Furthermore, germination arrest did not occur in absence of germination medium separating bacteria and seeds (*Figure 1—figure supplement 1C*).

From these observations, we conclude that *P. aeruginosa* releases a germination repressive activity (GRA) that diffuses in the germination medium.

### The GRA released by *P. aeruginosa* stimulates GA and ABA signaling pathways to repress seed germination

Upon imbibition seeds respond to unfavorable abiotic conditions by blocking GA synthesis, which promotes DELLA factors accumulation. In turn, DELLA factors collectively repress seed germination by stimulating ABA signaling including the accumulation of the germination repressor TF ABI5 (*Piskurewicz et al., 2008*, *2009*). We studied whether GA and ABA signaling pathways play a role to repress germination in seeds exposed to *P. aeruginosa*.

GA signaling mutant seeds, lacking one or several DELLA factors and particularly Δ*della* mutant seeds lacking all five DELLA factors, had a higher percentage of seed germination in presence of WT *P. aeruginosa* (*PAO1*) (*Figure 2A*). Furthermore, *Arabidopsis* mutant seeds deficient in ABA synthesis (*aba1*) or signaling (*abi3*, *abi5*) also had a higher percentage of seed germination in presence of *P. aeruginosa* (*PAO1*) (*Figure 2B*).

We also exposed seeds to lyophilized extracts of *P. aeruginosa* liquid culture medium (Materials and methods, *Figure 2—figure supplement 1A*). These bacteria-free extracts (thereafter referred as 'extracts') also elicited DELLA-dependent germination arrest responses (*Figure 2—figure supplement 1B, C and D*).

We monitored the accumulation of the DELLA factor RGL2 and the ABA response TF ABI5 in seeds exposed to *P. aeruginosa*. In absence of bacteria, RGL2 and ABI5 were detectable during the first 44 hr upon seed imbibition but their accumulation decreased thereafter, which was associated with increasing germination percentage, consistent with previous reports (*Figure 2C*; *Figure 2—*

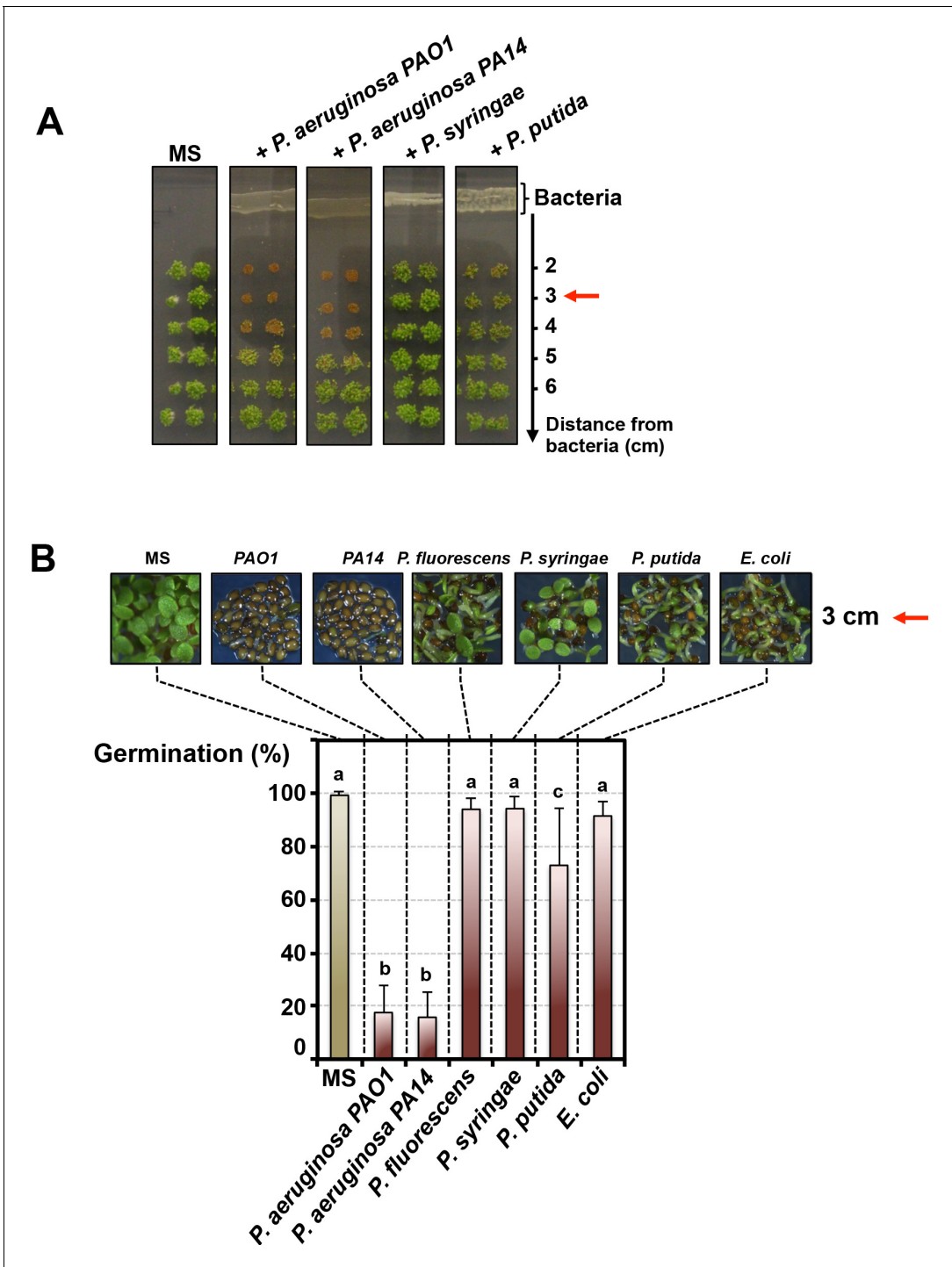

**Figure 1.** *Pseudomonas aeruginosa* releases a germination repressive activity (GRA). (**A**) Pictures show *Arabidopsis* plants 4 days after sowing WT seeds in germination medium lacking bacteria (MS) or containing a given *Pseudomonas* species as indicated. *Escherichia coli* was included as a Gram-negative non-*Pseudomonas* species control. Note that seeds in close proximity of *P. aeruginosa* strains mostly did not germinate. Red arrows indicate distance used to calculate germination percentage in B. (**B**) Same experiment as in A. Representative pictures of plants 4 days after sowing seeds in germination medium in absence (MS) or presence of bacteria as indicated. Histograms show seed germination percentage 4 days upon sowing seeds in absence (MS) or presence of bacteria as indicated. Data represent mean ± standard deviation (nine replicates, n = 300–350). Lower case letters above histograms are used to establish whether two seed germination percentage values are statistically significantly different as assessed by one-way ANOVA followed by a Tukey HSD test (p<0.05): different letters denote statistically different values.

DOI: https://doi.org/10.7554/eLife.37082.003

*Figure 1 continued on next page*

*Figure 1 continued*

The following source data and figure supplement are available for figure 1:

**Source data 1.** Germination percentage for each replicate.

DOI: https://doi.org/10.7554/eLife.37082.005

**Figure supplement 1.** *Pseudomonas aeruginosa* releases a diffusible GRA in the germination medium.

DOI: https://doi.org/10.7554/eLife.37082.004

*figure supplement 2A and B*) (*Lee et al., 2002*; *Piskurewicz et al., 2008*). However, WT seeds exposed to *P. aeruginosa* (*PAO1*) persistently accumulated RGL2 and ABI5 up to 148 hr upon imbibition, which was associated with low germination percentage (*Figure 2C*). In contrast, Δ*della* mutant seeds exposed to *P. aeruginosa* extracts did not maintain ABI5 accumulation over time upon imbibition, which was associated with increasing germination percentage (*Figure 2D*). These observations indicate that the GRA released by *P. aeruginosa* represses germination by promoting DELLA-dependent increase of the germination repressor ABI5, as previously shown with seeds unable to synthesize GA upon imbibition (*Piskurewicz et al., 2008*, *2009*).

## The GRA released by *P. aeruginosa* elicits DELLA-dependent responses in a GA-independent manner

However, and surprisingly, exogenous GA (10 μM) did not promote germination of WT seeds exposed to *P. aeruginosa* extracts (*Figure 3A*). This was associated with persistent accumulation of RGL2 and ABI (*Figure 3B*). In contrast, and expectedly, exogenous GA promoted the germination of WT seeds exposed to paclobutrazol (PAC), an inhibitor of GA synthesis, which was associated with downregulation of RGL2 and ABI5 protein levels, consistent with previous results (*Figure 3A*, *Figure 3—figure supplement 1* and *Figure 5B*). Thus, the GRA could elicit a DELLA-dependent germination arrest irrespective of GA levels in seeds.

To further study how the GRA triggers a DELLA-dependent germination arrest, we compared the early transcriptome of WT and Δ*della* seeds exposed to PAC or *P. aeruginosa* (*PAO1*) extract (Materials and methods, *Figure 3C and D*). We also included in the analysis the effect of exogenous GA (Materials and methods).

The expression of as many as 1103 genes was changed in PAC-treated WT seeds relative to untreated WT seeds (MS vs PAC, *Figure 3C*, *Supplementary file 1*). In contrast, only three genes had their expression affected in PAC-treated Δ*della* seeds relative to untreated Δ*della* seeds (MS vs PAC, *Figure 3C*, *Supplementary file 1*). Furthermore, only 36 and 2 genes had their expression changed when WT and Δ*della* seeds were treated with both PAC and GA (10 μM), respectively (MS vs PAC + GA, *Figure 3C*, *Supplementary file 1*). Thus, DELLA factors influence gene expression mainly when GA synthesis is compromised, consistent with previous reports (*Cao et al., 2006*).

In contrast to PAC-treated WT seeds, only 130 genes had their expression changed in WT seeds exposed to *PAO1* extract relative to non-exposed WT seeds (MS vs *PAO1* extract, *Figure 3D*, *Supplementary file 2*). Among the 130 genes, the expression of 87 genes was DELLA-dependent since it did not change in Δ*della* seeds exposed to *PAO1* extract (*Figure 3D*, *Supplementary file 2*). Strikingly, about a quarter of the 87 genes (24 genes) had their expression unchanged when GA was included in the medium together with *PAO1* extract (MS vs *PAO1* extract +GA, *Figure 3D*).

Altogether, these results show that the GRA released by *P. aeruginosa* is able to elicit gene expression responses in a DELLA-dependent and GA-independent manner.

## Release of the GRA by *P. aeruginosa* requires functional LASI/RHLI and IQS quorum sensing systems

We next sought to identify the GRA by first identifying *P. aeruginosa* genes that are necessary to its release.

Bacteria have a quorum sensing (QS) system regulating gene expression to mount coordinated behavioral responses when a bacterial population reaches high densities (*Ng and Bassler, 2009*). The *P. aeruginosa* QS has four QS subsystems influencing each other: 1) *las*, 2) *rhl*, 3) *pqs* and 4) the recently discovered IQS (for 'Integrating the QS network'). Each subsystem consists of genes or operons producing signaling molecules, referred as autoinducers, which interact with and activate

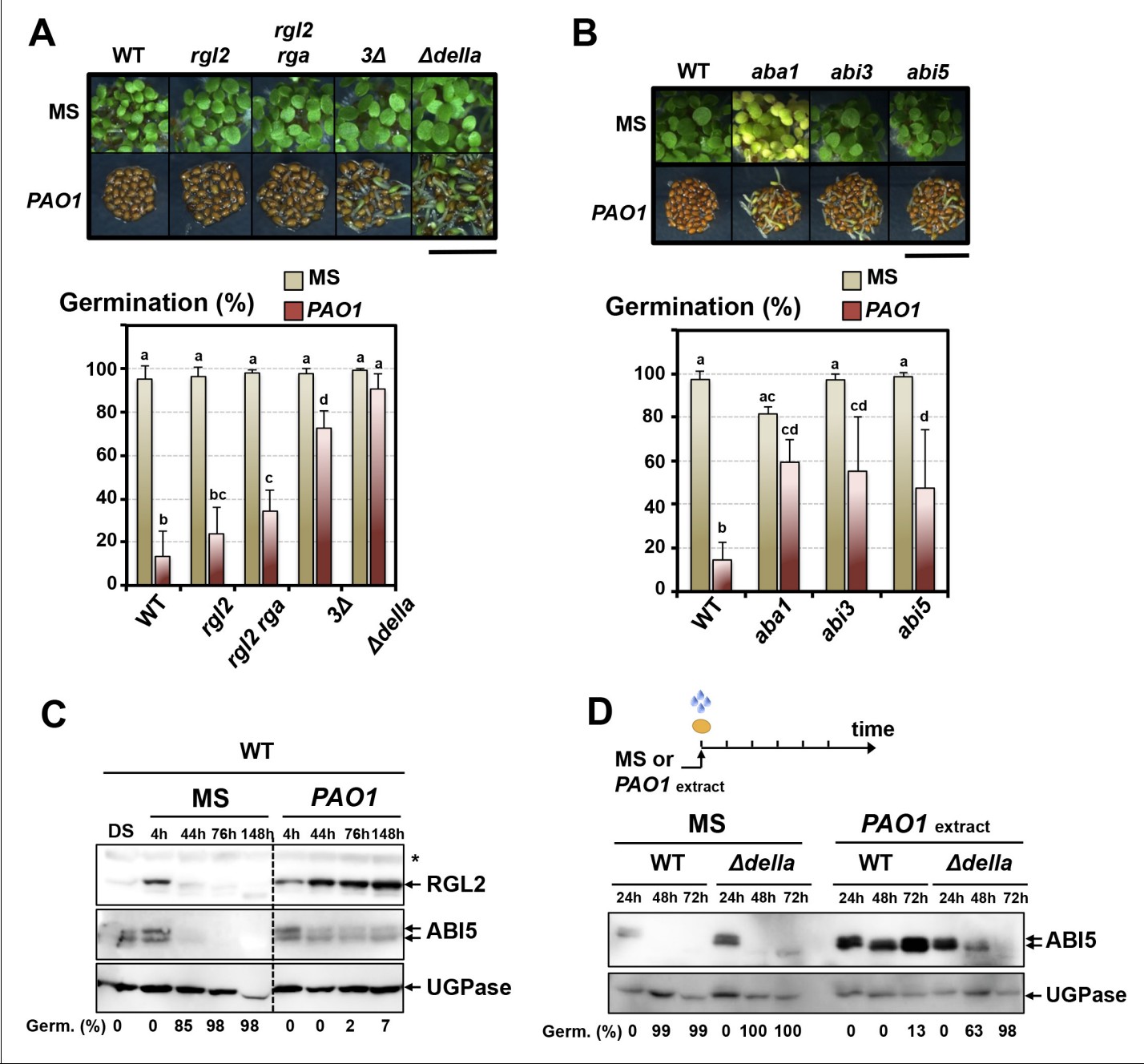

**Figure 2.** The GRA released by *P.aeruginosa* stimulates GA and ABA signaling pathways to repress seed germination. (**A**) Representative pictures of *Arabidopsis* plants 3 days after sowing WT and GA signaling mutant seeds in germination medium lacking bacteria (MS) or containing *P. aeruginosa* (*PAO1*). *Arabidopsis* plants include WT (Col-0) seeds and *rgl2*, *rgl2/rga*, *rgl2/rga/gai* (3Δ) and *rgl2/rga/gai/rgl1/rgl3* (Δ*della*) mutant seeds as indicated. Seeds and *P. aeruginosa* cells were separated by 2 cm. Histograms show seed germination percentage 3 days upon sowing seeds in absence (MS) or presence (*PAO1*) of bacteria as shown. Seeds and *P. aeruginosa* cells were separated by 2 cm. Data represent mean ± standard deviation (five replicates, n = 200–250). Statistical treatment and lower case letters as in *Figure 1B*. (**B**) Same as in A. using ABA synthesis and signaling mutant seeds. Mutants included *aba1* (deficient in ABA synthesis), *abi3* and *abi5* as indicated. Histograms show seed germination percentage 3 days upon sowing seeds in absence (MS) or presence (*PAO1*) of bacteria as indicated. Seeds and *P. aeruginosa* cells were separated by 2 cm. Data represent mean ± standard deviation (four replicates, n = 150–200). Statistical treatment and lower case letters as in *Figure 1B*. (**C**) Protein gel blot analysis of a time course of RGL2 and ABI5 protein levels upon WT (Col-0) seed imbibition in the absence (MS) or presence of *P. aeruginosa* (*PAO1*). Seeds and *P. aeruginosa* cells were separated by 2 cm. DS, dry seeds. UGPase protein levels are used as a loading control. Germination percentage at each time point is indicated. (**D**) Protein gel blot analysis of a time course of ABI5 protein levels upon WT (Col-0) and Δ*della* seed imbibition in the absence (MS)

*Figure 2 continued on next page*

*Figure 2 continued*

or presence of WT *P. aeruginosa* (*PAO1*) extract. UGPase protein levels are used as a loading control. Germination percentage at each time point is indicated. The asterisk (*) represents and unspecific banc detected by the RGL2 antibody.

DOI: https://doi.org/10.7554/eLife.37082.006

The following source data and figure supplements are available for figure 2:

**Source data 1.** Germination percentage for each replicate.

DOI: https://doi.org/10.7554/eLife.37082.009

**Figure supplement 1.** Lyophilized extracts of *P. aeruginosa* liquid culture medium elicit germination arrest responses similar to those observed with *P. aeruginosa* in germination plates.

DOI: https://doi.org/10.7554/eLife.37082.007

**Figure supplement 2.** RGL2, GAI, RGA and ABI5 antibodies specificity.

DOI: https://doi.org/10.7554/eLife.37082.008

cognate transcription factors controlling bacterial coordinated behavior (*Lee and Zhang, 2015*; *Papenfort and Bassler, 2016*; *Moradali et al., 2017*).

We asked whether the GRA released by *P. aeruginosa* is under QS control. We exposed WT seeds to *P. aeruginosa* mutant strains deficient in (1) the *LASI* and *RHLI* operons (Δ*lasl*Δ*rhll* double mutants), which control the production of the homoserine lactones autoinducers, (2) the *PQS* operon (Δ*pqsA* and Δ*pqsH* mutants), which controls the production of the quinolones autoinducers and (3) the *AMB* operon (Δ*ambE* mutant), which is necessary for the production of the autoinducer 2-(2-hydroxyphenyl)-thiazole-4-carbaldehyde (IQS) (*Lee et al., 2013a*).

We observed markedly lower GRAs released by Δ*lasl*Δ*rhll* and particularly with Δ*ambE* mutants (*Figure 4A*, *Figure 4—figure supplement 1*, *Figure 2—figure supplement 1C*). Lower GRAs were also observed with Δ*ambA-D* mutants, each affected in individual *AMB* operon genes (*Figure 4B*). Furthermore, and strikingly, transgenic WT *P. aeruginosa* bacteria overexpressing the *AMB* operon (*AMBox*) released a higher GRA relative to the parental *PAO1* line (*Figure 4—figure supplement 1*).

Altogether, these results show that the GRA released by *P. aeruginosa* is dependent on QS activity and particularly that of IQS. The importance of IQS is further suggested by the fact that *P. aeruginosa* is the only *Pseudomonas* species used in this study having the IQS QS (*Figure 1A*) (*Rojas Murcia et al., 2015*; *Lee et al., 2013b*).

## L-2-amino-4-methoxy-trans-3-butenoic acid (AMB) is the main GRA released by *P. aeruginosa*.

We undertook an unbiased metabolomic approach complemented with biochemical purification procedures to identify the GRAs present in *P. aeruginosa* extracts (*Figure 4—figure supplement 2*).

In the metabolomic approach, untargeted UHPLC-HRMS[2] (Ultra-High-Performance Chromatography hyphenated to High-Resolution Tandem Mass Spectrometry) data were acquired from extracts of *P. aeruginosa* strains releasing a GRA (*PAO1* and Δ*pqsa*), not releasing a GRA (Δ*ambE* and Δ*lasl/rhll*) or releasing a higher GRA (*AMBox*) (Materials and methods, *Figure 4—figure supplements 3,4*, *Supplementary file 3*). These analyses provided a list of biomarkers responsible for the metabolic differences between strains (*Supplementary file 4*). The top discriminant feature was the mass spectrometry signal corresponding to oxyvinylglycine L-2-amino-4-methoxy-trans-3-butenoic acid (also referred as methoxyvinylglycine or AMB) (*Figure 4—figure supplement 5*, *Supplementary file 3 and 4*).

We next used chromatographic methods to fractionate WT *PAO1*, *AMBox* and Δ*ambE* extracts, which led to the identification of a GRA present in a polar fraction of WT *PAO1* and *AMBox* extracts but absent in that of Δ*ambE* extracts (Materials and methods, *Figure 4—figure supplement 6A*). *AMBox* extracts were used to further fractionate the GRA-containing polar fraction (*Figure 4—figure supplement 6B*). This led to the isolation of a purified fraction containing AMB (as confirmed by NMR) (*Figure 4—figure supplement 6B*, *Appendix 1—figures 1* and *2*) and the GRA (*Figure 4—figure supplement 7*). AMB present in this purified fraction is hereafter referred as 'AMBi'.

Altogether, these results point to AMB as being the main GRA.

Consistent with this hypothesis synthetic AMB dose response for germination inhibition confirmed that AMB inhibits WT seed germination (*Figure 4C*). Furthermore, mutant seeds deficient in GA

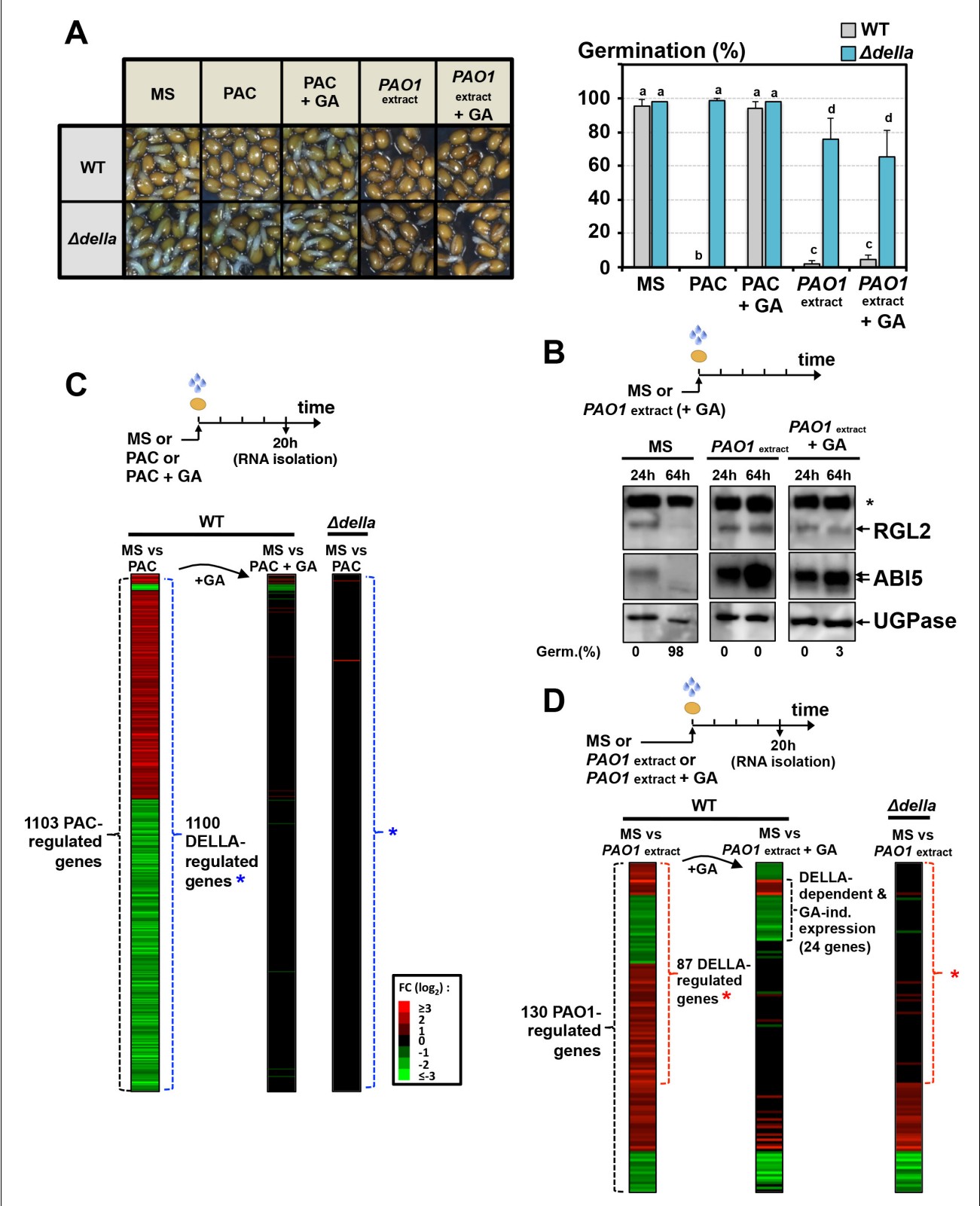

**Figure 3.** The GRA released by *P. aeruginosa* elicits DELLA-dependent responses in a GA-independent manner. (**A**) Representative pictures of *Arabidopsis* plants 48 hr after sowing WT and *Δdella* seeds in absence (MS) or presence of 5 µM paclobutrazol (PAC) or 5 µM PAC and 10 µM GA (PAC + GA) or WT *P. aeruginosa* extract without (*PAO1* extract) or with 10 µM GA (*PAO1* extract + GA) as indicated. 0.7 mg/ml of WT *P. aeruginosa* extract was used. Histograms show seed germination percentage 48 hr upon sowing seeds under conditions as indicated. Data represent

*Figure 3 continued on next page*

*Figure 3 continued*

mean ± standard deviation (three replicates, n = 150–200). Statistical treatment and lower case letters as in *Figure 1B*. (B) Protein gel blot analysis of RGL2 and ABI5 levels in WT (Col) seeds 24 hr and 64 hr after imbibition in the absence (MS) or presence of WT *P. aeruginosa* extract without (*PAO1* extract) or with 10 µM GA (*PAO1*extract + GA) as indicated. UGPase protein levels are used as a loading control. Germination percentage at each time point is indicated. The asterisk (*) represents and unspecific banc detected by the RGL2 antibody. (C) Diagram describes the procedure to isolate seed RNA in order to compare early transcriptomes of WT and Δ*della* seed exposed to PAC or PAC with GA. Total RNA isolated from WT (Col-0) and Δ*della* seeds imbibed for 20 hr in the absence (MS) or presence of 5 µM paclobutrazol without (PAC) or with 10 µM GA (PAC + GA) was used for RNAseq analysis (two replicates). The red and green horizontal lines represent individual genes whose mRNA expression is significantly upregulated (red) or downregulated (green) by at least twofold under different seed treatments as follows: left column represents the 1103 genes whose expression is either upregulated or downregulated in PAC-treated seeds relative to non-treated seeds (MS) (blue asterisk indicates the genes whose expression does not change in PAC-treated Δ*della* seeds); middle column represents the expression of the 1103 genes in seeds treated with PAC and GA (PAC + GA) relative to non-treated WT seeds (MS); right column represents the expression of the 1103 genes in Δ*della* seeds treated with PAC relative to non-treated seeds (MS). The scale bar relates color with absolute fold changes. Black color represents no change in gene expression. (D) Same experiment as in C using WT and Δ*della* seed exposed to WT *P. aeruginosa* extract without (*PAO1* extract) or with 10 µM GA (*PAO1*extract + GA) as indicated. 0.7 mg/ml of WT *P. aeruginosa* extract was used. Red asterisk indicates the genes whose expression does not change in Δ*della* seeds treated with *PAO1* extract.

DOI: https://doi.org/10.7554/eLife.37082.010

The following source data and figure supplement are available for figure 3:

**Source data 1.** Germination percentage for each replicate.
DOI: https://doi.org/10.7554/eLife.37082.012

**Figure supplement 1.** Effect of exogenous GA on RGL2 and ABI5 protein level under low GA condition.
DOI: https://doi.org/10.7554/eLife.37082.011

signaling (Δ*della*), ABA biosynthesis (*aba1*) or ABA signaling (*abi3,*) had a higher percentage of seed germination in presence of synthetic AMB (*Figure 4D*).

A concentration of 50 µM synthetic AMB had a GRA equivalent to that 0.8 mg/ml *PAO1* extract in germination plates (see dashed red line in *Figure 4C E*). However, we estimated that this extract concentration provides only 16 µM of natural AMB (quantification by targeted UHPLC-HRMS[2] approach, *Figure 4—figure supplement 8*). This discrepancy could be explained as follows: (1) there are other compounds present in *PAO1* extracts enhancing the GRA of AMB, (2) AMB is not the only compound released by *P. aeruginosa* having a substantial GRA. To test the first possibility, we supplemented Δ*ambE* extracts, lacking AMB, with synthetic AMB so as to match the amount of AMB naturally present in *PAO1* extracts. The GRA of the AMB-supplemented Δ*ambE* extract was similar to that present in *PAO1* extracts (*Figure 4E*).

Furthermore, we also purified the natural AMB present in *Pseudomonas* extracts (Materials and methods, *Figure 4—figure supplement 6B*). Δ*ambE* extracts supplemented with AMBi or synthetic AMB contained the same GRA (*Figure 4—figure supplement 9*).

Taken together, these results strongly suggest that the GRA of synthetic AMB can be enhanced by the presence of other compounds present in *P. aeruginosa* extracts.

We also tested the GRA of aminoethoxyvinylglycine (AVG), another oxyvinylglycine. As much as 200 µM of AVG of did not noticeably repress *Arabidopsis* seed germination, consistent with previous reports, indicating that AMB is an oxyvinylglycine specifically repressing *Arabidopsis* seed germination (*Figure 4—figure supplement 10*) (*Wilson et al., 2014*). AVG is known to inhibit ethylene synthesis, suggesting that AMB does not block germination by inhibiting ethylene synthesis (further discussed below) (*Adams and Yang, 1979*; *Huai et al., 2001*; *Capitani et al., 2002*).

We next verified whether AMB could recapitulate the effects of *P. aeruginosa* cells or WT *PAO1* extracts on RGL2 and ABI5 protein accumulation. WT seeds exposed to AMB persistently accumulated RGL2 and ABI5 over time, unlike unexposed seeds (*Figure 5A*). Persistent high ABI5 accumulation was not observed in AMB-treated Δ*della* mutant seeds (*Figure 5A*). Furthermore, beyond 24 hr of imbibition in presence of AMB, PAC and GA, RGL2 protein levels persisted in seeds despite the presence of GA, unlike seeds treated with PAC and GA only (*Figure 5B*). This was associated with persistent ABI5 accumulation and absence of seed germination (*Figure 5B*). Thus, AMB induces changes in RGL2 and ABI5 accumulation in a manner similar to that observed with *P. aeruginosa* or WT PAO1 extracts (*Figures 2C, D* and *3B*).

Altogether, these results conclusively show that AMB is the main GRA released by *P. aeruginosa* repressing germination in a DELLA- and ABA-signaling-dependent manner.

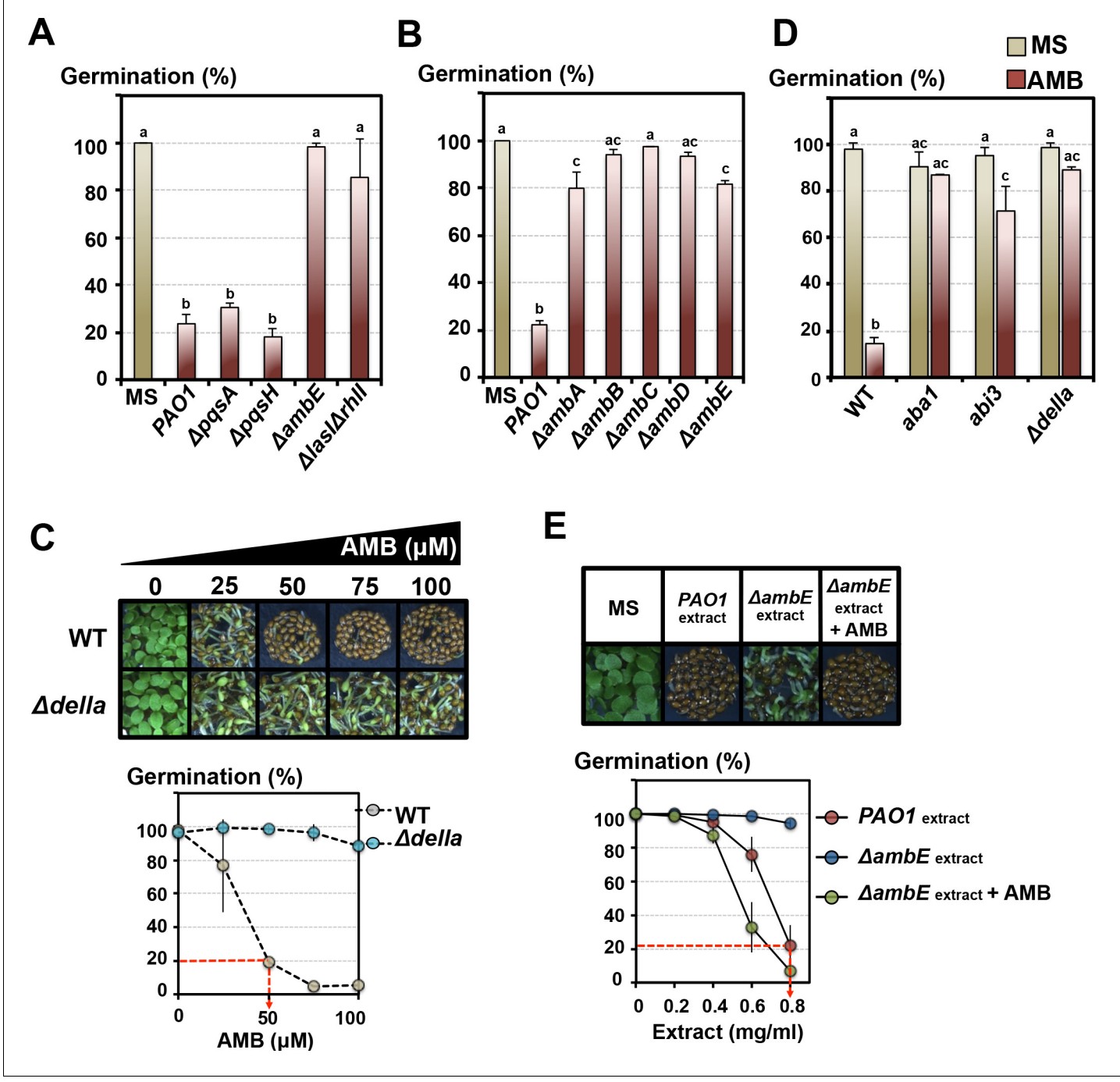

**Figure 4.** The GRA is under the control of the *P. aeruginosa* LASI/RHLI and IQS QS systems. L-2-amino-4-methoxy-trans-3-butenoic acid (AMB) is the main GRA released by *P. aeruginosa*. (A) Histograms show seed germination percentage 4 days upon sowing seeds in absence (MS) or presence of WT *P. aeruginosa* (*PAO1*) or mutant *P. aeruginosa* strains as indicated. *PAO1* is the reference WT *P. aeruginosa* strain from which all the QS mutant strains were derived. Seeds and *P. aeruginosa* cells were separated by 2 cm. Data represent mean ± standard deviation (four replicates, n = 150–200). Statistical treatment and lower case letters as in *Figure 1B*. (B) Same experiment as in A. using *P. aeruginosa* Δ*ambA-E* mutants, as indicated, each affected in individual *AMB* operon genes. Data represent mean ± standard deviation (three replicates, n = 100–150). Statistical treatment and lower case letters as in *Figure 1B*. (C) AMB represses *Arabidopsis* seed germination in a DELLA-dependent manner. Pictures show representative *Arabidopsis* plants 4 days after sowing WT and Δ*della* seeds in presence of different AMB concentrations as indicated. The graph shows quantification of the germination percentage of WT and Δ*della* seeds exposed to AMB. The red dashed line indicates the concentration of synthetic AMB (50 µM) having the same GRA as that present in germination plates containing 0.8 mg/ml of *PAO1* extracts (see *Figure 4E*). Data represent mean ± standard deviation (three replicates, n = 150–200). (D) AMB represses *Arabidopsis* seed germination in an ABA-dependent manner. Histograms show

*Figure 4 continued on next page*

*Figure 4 continued*

germination percentage of WT, *aba1*, *abi3* and *Δdella* 3 days after sowing seeds in absence (MS) or presence of 50 μM of synthetic AMB. Data represent mean ± standard deviation (two replicates, n = 100–150). Statistical treatment and lower case letters as in *Figure 1B*. (E) Synthetic AMB introduces a GRA in *ΔambE* extracts equivalent to that of *PAO1* extracts. *ΔambE* extracts, which lack AMB, were supplemented with synthetic AMB so as to obtain the same amount of AMB naturally present in *PAO1* extracts. Pictures show representative *Arabidopsis* plants 4 days after sowing WT seeds in absence (MS) or presence of WT *P. aeruginosa* extracts (*PAO1*extract) or *ΔambE P. aeruginosa* extracts (*ΔambE*extract) or AMB-supplemented *ΔambE* extracts (*ΔambE*extract + AMB). 0.8 mg/ml of WT *P. aeruginosa* extract was used. The graph shows quantification of the germination percentage of WT seeds exposed to the different extract concentrations as indicated. Note that the supplemented *ΔambE* extract (*ΔambE*extract + AMB) has a GRA equivalent, if not higher, than that of *PAO1* extract. The red dashed line indicates the concentration of *PAO1* extract (0.8 mg/ml) having the same GRA as that present in germination plates containing 50 μM synthetic AMB (see *Figure 4C*). Data represent mean ± standard deviation (six replicates, n = 250–300).
DOI: https://doi.org/10.7554/eLife.37082.013

The following source data and figure supplements are available for figure 4:

**Source data 1.** Germination percentage for each replicate.
DOI: https://doi.org/10.7554/eLife.37082.025
**Source data 2.** Germination percentage for each replicate.
DOI: https://doi.org/10.7554/eLife.37082.026
**Figure supplement 1.** The GRA is under the control of the *P. aeruginosa* LASI/RHLI and IQS QS systems.
DOI: https://doi.org/10.7554/eLife.37082.014
**Figure supplement 2.** Approaches used in this study to identify compounds released by *P. aeruginosa* with a germination repressive activity (GRA).
DOI: https://doi.org/10.7554/eLife.37082.015
**Figure supplement 3.** Unsupervised and OPLS-DA results from metabolomic analysis 1.
DOI: https://doi.org/10.7554/eLife.37082.016
**Figure supplement 4.** Unsupervised and OPLS-DA results from metabolomic analysis 2.
DOI: https://doi.org/10.7554/eLife.37082.017
**Figure supplement 5.** *m/z* 327.12 is the Mass Spectrometry (MS) signature of AMB.
DOI: https://doi.org/10.7554/eLife.37082.018
**Figure supplement 6.** Isolation of the GRA released by *P.aeruginosa.*
DOI: https://doi.org/10.7554/eLife.37082.019
**Figure supplement 7.** Fraction F37 contains AMB and a GRA.
DOI: https://doi.org/10.7554/eLife.37082.020
**Figure supplement 8.** Quantification of AMB in *Pseudomonas* extracts.
DOI: https://doi.org/10.7554/eLife.37082.021
**Figure supplement 9.** *ΔambE* extracts supplemented with AMBi or synthetic AMB contain the same GRA.
DOI: https://doi.org/10.7554/eLife.37082.022
**Figure supplement 10.** The oxyvinylglycine AVG does not inhibit *Arabidopsis* seed germination.
DOI: https://doi.org/10.7554/eLife.37082.023
**Figure supplement 11.** Presence of the *ambABCDE* operon in at least 15 other *P. aeruginosa* strains.
DOI: https://doi.org/10.7554/eLife.37082.024
**Figure supplement 11—source data 1.** Description of 15 different Pseudomonas aeruginosa strains containing the ambABCDE operon in their genomes.
DOI: https://doi.org/10.7554/eLife.37082.027

## Genetic experiments indicate that DELLA factor activity to stimulate ABA signaling is enhanced in AMB-treated seeds

To better understand the genetic requirement of functional DELLA genes for the AMB-dependent germination arrest, we focused on the DELLA factors RGL2, GAI and RGA (*Piskurewicz et al., 2008*; *2009*).

We asked whether GA-dependent DELLA protein degradation was perturbed in AMB-treated seeds. WT seeds were imbibed under normal (MS) conditions or in presence of PAC, AMB or ABA. As expected, RGL2, GAI and RGA protein levels markedly increased in presence of PAC, consistent with the notion that low GA stabilizes DELLA proteins (*Figure 6A, B and C*, *Figure 2—figure supplement 2C*). In contrast, no marked increase in RGL2, GAI and RGA protein levels was observed in presence of AMB or ABA despite the fact that both treatments arrested seed germination as in PAC-treated seeds (*Figure 6A, B and C*).

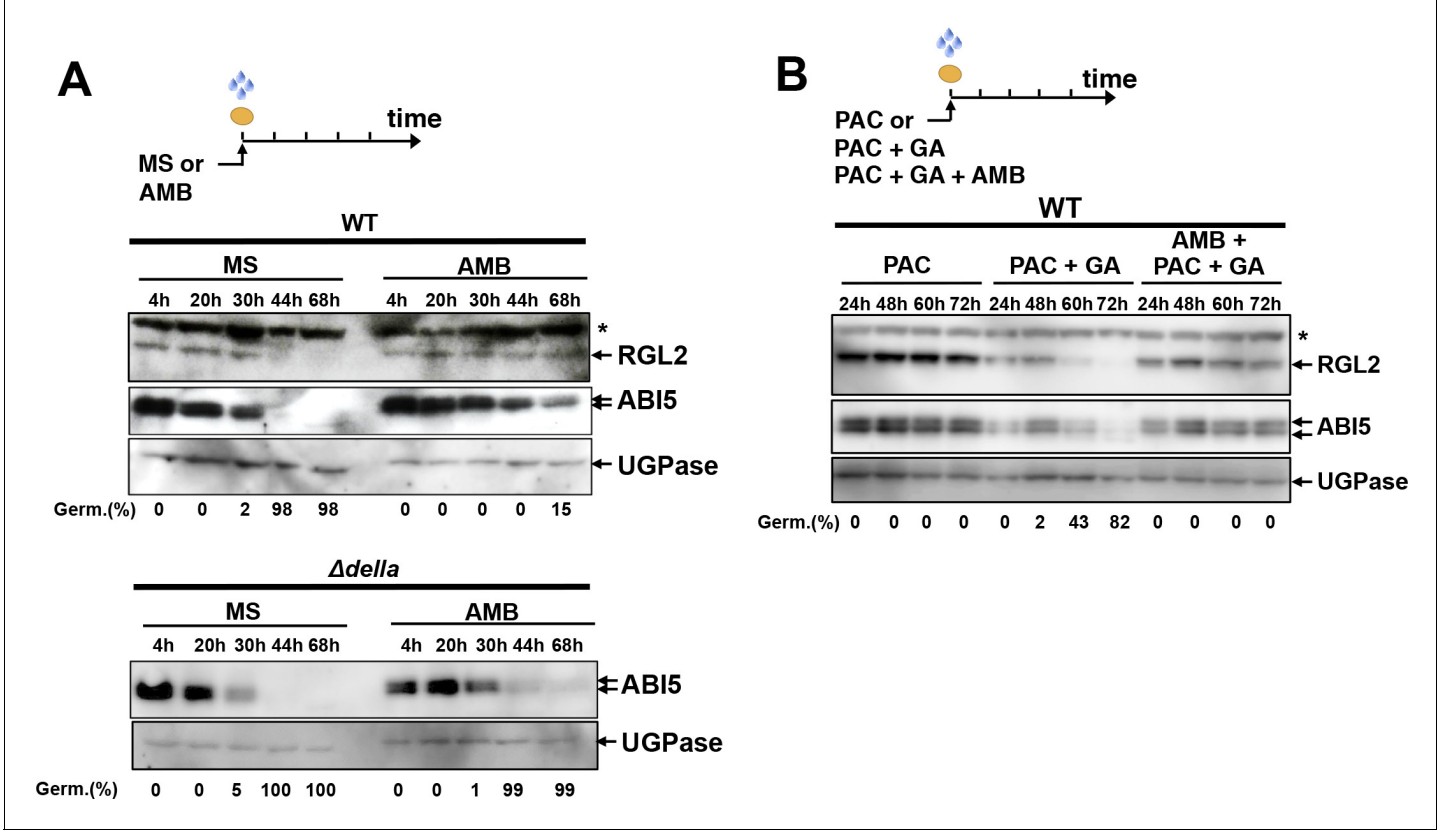

**Figure 5.** AMB induces changes in RGL2 and ABI5 accumulation in a manner similar to that observed with *P. aeruginosa* or WT *PAO1* extracts. (**A**) Protein gel blot analysis of a time course of RGL2 and ABI5 protein levels upon seed imbibition in the absence (MS) or presence of 50 µM AMB (AMB). UGPase protein levels are used as a loading control. Top panel: RGL2 and ABI5 protein levels in WT seeds. Bottom panel: ABI5 protein levels in Δ*della* seeds. Germination percentage at each time point is indicated. The asterisk (*) represents and unspecific banc detected by the RGL2 antibody. (**B**) Same as in A using seeds imbibed in the presence of 10 µM PAC, 1 µM GA or 100 µM AMB as indicated. Germination percentage at each time point is indicated.

DOI: https://doi.org/10.7554/eLife.37082.028

We also monitored the extinction of RGL2, GAI and RGA protein accumulation upon exposure to GA in absence or presence of AMB (*Figure 6D*, *Figure 6—figure supplement 1A and B*). WT seeds were first treated for 30 hr with PAC to allow for DELLA protein stabilization and high accumulation. After 30 hr seeds were further treated with AMB for 12 hr to ensure AMB presence within seed tissues prior to adding GA. Upon addition of GA to the medium RGL2, RGA and GAI protein levels decreased over 28 hr in a similar manner to that observed in control plates lacking AMB (*Figure 6D*, *Figure 6—figure supplement 1A and B*). Penetrance of AMB within seeds was confirmed by the absence of seed germination of AMB-treated seeds despite the presence of GA (*Figure 6D*).

Altogether, these results strongly suggest that GA-dependent DELLA protein degradation is not affected in AMB-treated seeds.

RGL2 persistently accumulated at late time points upon seed imbibition in presence of AMB despite the presence of GA (*Figure 5B*). Furthermore, ABA signaling in seeds promotes *RGL2* mRNA accumulation as well as that of *ABI5* (*Lopez-Molina et al., 2001*; *Piskurewicz et al., 2008*; *2009*). We therefore hypothesized that the DELLA activity promoting ABA signaling in seeds is enhanced in AMB-treated seeds (*Lopez-Molina et al., 2001*; *Piskurewicz et al., 2008*; *2009*). In turn, this would explain the persistent RGL2 and ABI5 protein accumulation in AMB-treated seeds (*Figures 2C,4A B*).

To test this hypothesis genetically, we measured *RGL2* and *ABI5* mRNA levels using total RNA isolated 30 hr and 44 hr upon imbibition from the same seed material used in *Figure 5A*. We also analyzed the expression of the ABA-responsive genes *EM1* and *NCED6* (*Lopez-Molina and Chua,*

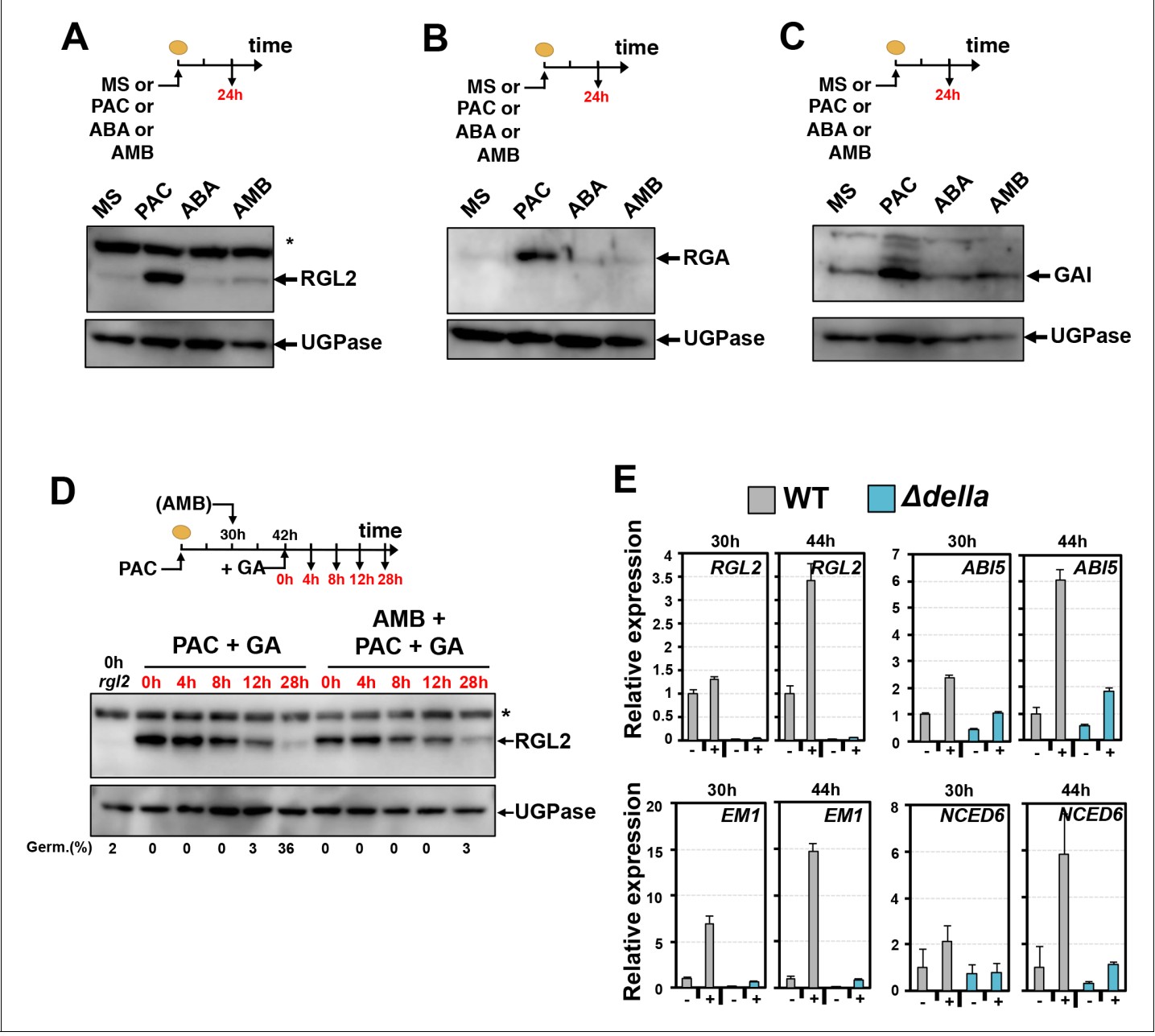

**Figure 6.** AMB and PAC induce different changes on DELLA protein accumulation. AMB promotes DELLA factor activity to stimulate ABA signaling. (A-C) Protein gel blot analysis using antibodies to RGL2 (A), RGA (B) and GAI (C) as indicated. Protein extracts from WT (Col seeds harvested 24 hr after seed imbibition in absence (MS) or presence of 10 µM PAC (PAC), 5 µM ABA (ABA) or 50 µM AMB (AMB). UGPase protein levels are used as a loading control. (D) AMB does not interfere with GA-dependent RGL2 protein downregulation. WT seeds are imbibed in presence of 5 µM PAC for 30 hr to trigger high RGL2 accumulation. Thereafter, seeds are transferred to germination plates containing 5 µM PAC or containing 5 µM PAC and 50 µM AMB. After 12 hr, 1 µM GA is further added and RGL2 protein levels are followed by protein gel blot analysis over the time points indicated in red. UGPase protein levels are used as a loading control. Germination percentage at each time point is indicated. The asterisk (*) represents and unspecific banc detected by the RGL2 antibody. (E) AMB stimulates DELLA activity to promote ABA-dependent responses. Histograms show *RGL2, ABI5, EM1* and *NCED6* mRNA accumulation in WT (Col) and Δ*della* seeds treated as described in A. For each time point, mRNA levels are normalized to mRNA levels in WT seeds sown in absence of AMB. Data represent mean ±standard deviation (three replicates).

DOI: https://doi.org/10.7554/eLife.37082.029

The following figure supplement is available for figure 6:

**Figure supplement 1.** AMB does not interfere with GA-dependent GAI and RGA protein downregulation.

DOI: https://doi.org/10.7554/eLife.37082.030

*2000*; *Lopez-Molina et al., 2001*; *Lefebvre et al., 2006*; *Martínez-Andújar et al., 2011*). Upon 30 hr of imbibition, seed accumulated the same RGL2 levels in absence (MS) or presence of AMB (AMB) (*Figure 5A*). Nevertheless, AMB-treated seeds accumulated markedly higher *ABI5*, *EM1* and *NCED6* mRNA levels relative to seeds imbibed in absence of AMB (*Figure 6E*). In contrast, AMB-treated Δ*della* mutant seeds accumulated lower *ABI5*, *EM1* and *NCED6* mRNA expression at 30 hr (*Figure 6E*). Upon 44 hr of imbibition, AMB-treated seeds further increased the expression of *ABI5*, *EM1* and *NCED6* mRNA as well as that of *RGL2* (*Figure 6E*).

Collectively, these genetic observations support the notion that the DELLA activity promoting ABA signaling in seeds is enhanced in AMB-treated seeds (*Figure 7E*). Whether this is the result of a direct interaction between AMB and DELLAs is not known.

Interestingly, AMB could still mildly stimulate *EM1*, *NCED6* and particularly *ABI5* mRNA expression in Δ*della* mutant seeds (*Figure 6E*). Accordingly, ABI5 protein accumulation, although diminishing, remained higher 44 hr and 68 hr after imbibition of AMB-treated Δ*della* seeds relative to non-treated seeds (*Figure 5A*). These genetic observations suggest that a residual activity promoting ABA synthesis or signaling independently of DELLA factors is present in AMB-treated seeds.

## Germination arrested seeds are protected from developmental abnormalities triggered by AMB

The relative proportion of germinating seeds relative to non-germinating seeds depended on the distance separating seeds from *P. aeruginosa* cells or on the concentration of AMB used in the germination plate (*Figure 1A*, *Figure 1—figure supplement 1*). Seeds that germinated in either presence of *P. aeruginosa* or AMB similarly produced pale seedlings whose growth was severely delayed bearing diminutive roots (*Figure 7A*). These developmental defects diminished with increased distance between seeds and bacteria or decreased AMB concentrations. Furthermore, they were no longer observed in seeds exposed to Δ*ambE P. aeruginosa* mutants, unable to produce AMB (*Figure 7A*). Altogether, these observations show that AMB released by *P. aeruginosa* severely perturbs seedling development.

Remarkably, within a seed population exposed to *P. aeruginosa,* WT seeds that did not germinate germinated upon transfer to plates lacking *P. aeruginosa* and produced normal seedlings (*Figure 7B*). The resulting WT seedlings produced a normal seed yield (*Figure 7C*). In contrast, WT seeds that had germinated in presence of *P. aeruginosa* failed to recover (*Figure 7B*). Furthermore, when WT and Δ*della* seeds were exposed for 3 days to *P. aeruginosa* prior to transfer to a bacteria-free medium, WT seeds had a higher survival rate relative to Δ*della* seeds (*Figure 7D*).

These observations therefore strongly suggest that germination-arrested seeds in presence of AMB are able to retain their vitality unlike newly emerged seedlings.

## Discussion

Whether biotic factors present in the soil affect seed germination responses is poorly understood. Here we show that *Arabidopsis* seeds respond when exposed to *P. aeruginosa* by blocking their germination. This involves perception of the oxyvinylglycine AMB released by *P. aeruginosa*, which triggers a DELLA-dependent seed germination arrest.

### Oxyviniglycines differentially affect seed germination

Oxyvinylglycines are a class bacterially produced compounds whose biological function is unclear. Oxyvinylglycines are known to inhibit irreversibly pyridoxal phosphate (PLP)-dependent enzymes (*Berkowitz et al., 2006*). Oxyvinylglycines had been previously associated with germination repressive activities (GRAs) after exposing graminaceous (Poaceae) seeds to bacterial culture filtrates. The best documented cases are those of AVG (aminoethoxyvinylglycine, produced by *Streptomyces sp*) and FVG (4-formylaminooxyvinylglycine, produced by *Pseudomonas fluorescens strain WH6*) that were directly and indirectly linked with a GRA, respectively (*McPhail et al., 2010*; *Okrent et al., 2017*). Only in the case of AVG a GRA could be established using a pure synthetic compound: 100 μM AVG inhibited the germination of *Poa annua* seeds. However, we found here that as much as 200 μM AVG did not inhibit *Arabidopsis* seed germination, consistent with previous reports (*Wilson et al., 2014*). More recently, using *Poa* seeds, Lee *et al*. also found a weak GRA in culture filtrates from *P. aeruginosa* strains overexpressing the *AMB* operon (*AMBox*) relative to that of AVG

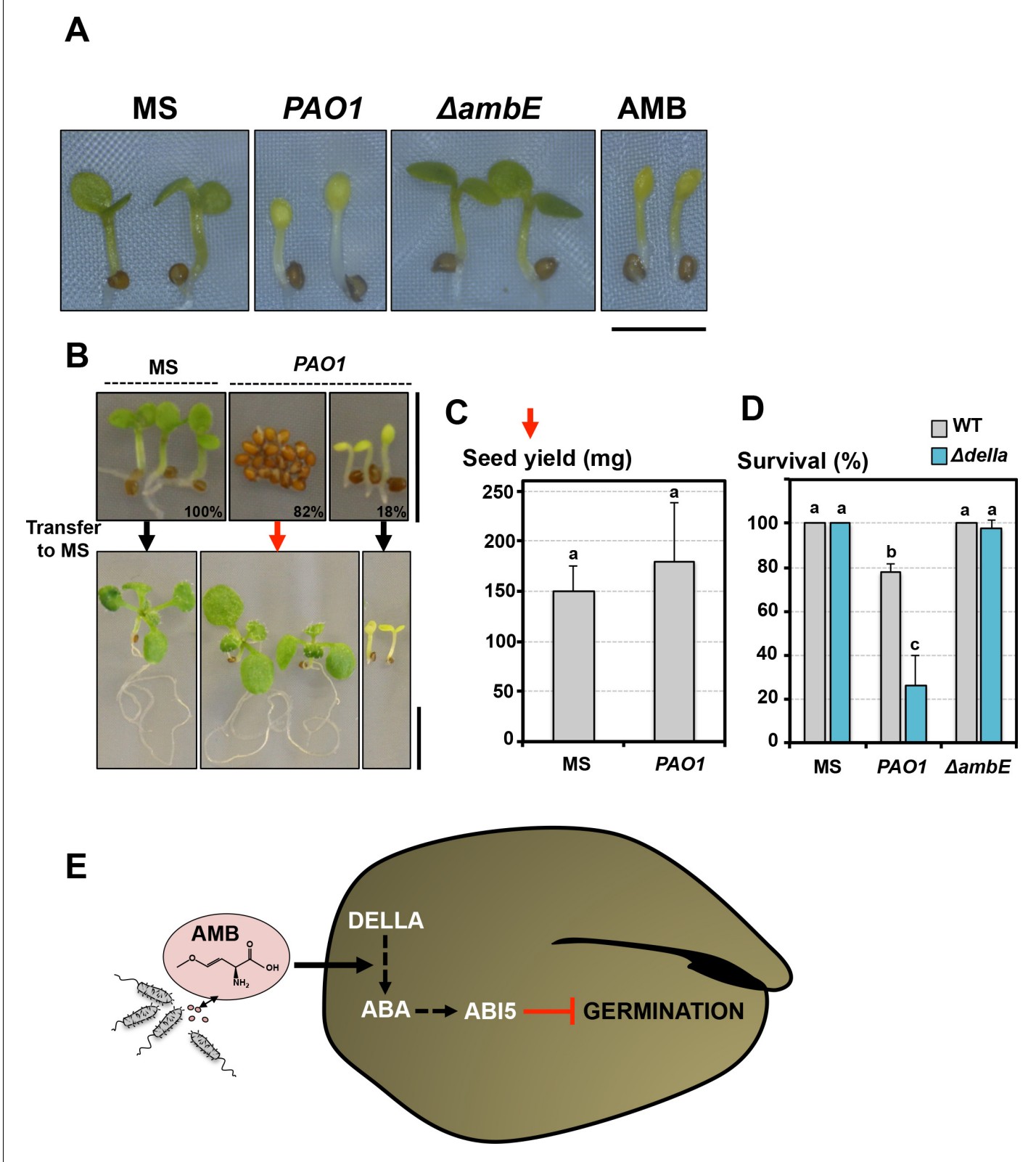

**Figure 7.** Germination-arrested seeds are protected from developmental abnormalities triggered by AMB. (**A**) Representative pictures of 4-day-old seedlings produced by seeds that germinated in absence (MS) or presence of WT *P. aeruginosa* (*PAO1*) or Δ*ambE* mutant *P. aeruginosa* (Δ*ambE*) or 50 μM AMB (AMB). Seeds and bacteria were separated by 2 cm. Scale bar: 2 mm. (**B**) Seeds were sown in absence (MS) or presence of *P. aeruginosa* (*PAO1*). Seeds and bacteria were separated by 2 cm. Top panel: representative pictures of plants 3 days after sowing. All the seeds sowed in absence

*Figure 7 continued on next page*

*Figure 7 continued*

of *P.aeruginosa* germinated 3 days after sowing (100%). Plants produced by seeds sown in presence of *P. aeruginosa* are shown according to their germination status 3 days after sowing: 82% (82%) did not germinate and 18% (18%) germinated. 3 days after sowing, plant material was transferred to a medium lacking *P. aeruginosa* (MS) and cultured for 1 week. Bottom panel shows representative pictures of plants 1 week after the transfer. (**C**) Histograms show the seed yield (expressed in mg) from plants produced by seeds never exposed to *P. aeruginosa* (MS) or from plants produced by seeds that did not germinate for 3 days in presence of *P. aeruginosa* and then transferred to medium lacking *P. aeruginosa*. Average seed yield was calculated from seeds produced by five and nine individual plants from seeds sown in absence or presence of *P. aeruginosa*, respectively. Data represent mean ± standard deviation. Statistical treatment and lower case letters as in *Figure 1B*. (**D**) WT and Δ*della* seeds were exposed for three days to WT *P. aeruginosa* (*PAO1*) prior to transfer to a bacteria-free medium. Histograms show percent of survival one week after transfer. Data represent mean ± standard deviation (three replicates, n = 60). Statistical treatment and lower case letters as in *Figure 1B*. (**E**) Model describing AMB mode of action in seeds.

DOI: https://doi.org/10.7554/eLife.37082.031

The following source data and figure supplements are available for figure 7:

**Source data 1.** Seed yield per plant (mg).

DOI: https://doi.org/10.7554/eLife.37082.038

**Source data 2.** Survival percentage.

DOI: https://doi.org/10.7554/eLife.37082.039

**Figure supplement 1.** Effect of WT *PAO1* extracts, Δ*ambE* extracts and 250 µM AVG on the germination of various brassicaceae seeds, as indicated (A to H).

DOI: https://doi.org/10.7554/eLife.37082.032

**Figure supplement 2.** Arrested brassicaceae seeds germinate after transfer to MS.

DOI: https://doi.org/10.7554/eLife.37082.033

**Figure supplement 3.** Representative pictures of brassicaceae seedlings in response to *PAO1* or Δ*ambE* extract.

DOI: https://doi.org/10.7554/eLife.37082.034

**Figure supplement 4.** Effect of AgNO3, *PAO1* and Δ*ambE* extracts on seed germination.

DOI: https://doi.org/10.7554/eLife.37082.035

**Figure supplement 5.** Effect of Yucasin, *PAO1* and Δ*ambE* extracts on seed germination.

DOI: https://doi.org/10.7554/eLife.37082.036

**Figure supplement 6.** Effect of PAG, methionine and *PAO1* extracts on seed germination.

DOI: https://doi.org/10.7554/eLife.37082.037

and FVG (*Lee et al., 2013b*). However, the concentrations of AMB or FVG in the germination assays were unspecified and no synthetic compound was used to directly test their intrinsic GRA. Furthermore, dicot seeds are less responsive to *P. fluorescens* culture filtrates than graminaceous monocot seed, suggesting that FVG is less active to repress the germination of dicot seeds including seeds of cabbage, which is a brassicaceae (*Banowetz et al., 2008*). We found that AMB-containing WT *PAO1* extracts inhibited the germination of brassicaceae seeds, including cabbage seeds, unlike Δ*ambE* extracts (*Figure 7—figure supplement 1*). No such effect was observed with 250 µM AVG (*Figure 7—figure supplement 1*). Arrested seeds germinated upon transfer to MS, indicating that the germination arrest is not due to a toxic effect (*Figure 7—figure supplement 2*).

Altogether, these results indicate that oxyvinylglycines do not affect seed germination in the same manner. They also strongly suggest that AMB is an oxyvinylglycine able to repress the germination of several seed dicot species.

Furthermore, brassicaceae seeds that germinated in presence of WT *PAO1* extracts developed developmental defects unlike those germinated in presence of Δ*ambE* extracts (*Figure 7A*, *Figure 7—figure supplement 3*). Germination arrested *Capsella* seeds in presence of WT *PAO1* extracts were protected as they produced normal seedlings upon transfer to a normal medium (*Figure 7—figure supplement 3E and F*).

Since oxyvinylglycines inhibit irreversibly pyridoxal phosphate (PLP)-dependent enzymes their proposed GRA was interpreted as a toxic effect killing the seed, rendering these compounds potentially useful as herbicides (*Rando, 1974*; *Lee et al., 2013b*). However, whether oxyvinylglycines repress seed germination in a manner requiring functional DELLA genes, as shown here, was not previously investigated. It should be noted that a signaling role for oxyvinylglycines is not incompatible with toxic effects at high concentrations as in the case of auxin whose synthetic derivative 2,4-D is widely used as a systemic herbicide (*Grossmann, 2007*).

AVG is a well-known inhibitor of 1-aminocyclopropane-1-carboxylic acid (ACC) synthase, a PLP-dependent enzyme catalyzing the synthesis of ACC, a precursor of ethylene, from S-adenosylmethionine (*Adams and Yang, 1979*; *Huai et al., 2001*; *Capitani et al., 2002*). AVG was also shown to inhibit auxin synthesis (*Soeno et al., 2010*). AMB may also inhibit ACC synthase as suggested by a report showing that AMB decreases ethylene levels in apples; however, whether AMB can inhibit ACC synthase in vitro was not shown (*Mattoo et al., 1979*).

Wilson *et al.* previously reported that 5 µM AVG, a widely used inhibitor of ACC synthase, lowers ethylene synthesis in *Arabidopsis* seeds and does not inhibit germination (*Wilson et al., 2014*). Here, we report that as much as 200 µM AVG did not noticeably inhibit seed germination, further confirming the results of Wilson et al. (*Figure 4—figure supplement 10*). Furthermore, heptuple *acs* mutant seeds, deficient several in *ACS* biosynthetic genes, germinated similarly to WT seeds (*Figure 7—figure supplement 4A and B*). Seeds deficient in *ACS* biosynthetic genes would be expected to respond more strongly to AMB-containing WT *PAO1* extracts if AMB represses germination by inhibiting ethylene biosynthesis. However, this is not what we observed (*Figure 7—figure supplement 4A and B*).

We also treated seeds with silver nitrate (AgNO3), which induces ethylene insensitivity, and did not observe an inhibition of seed germination (*Figure 7—figure supplement 4C*) (*Rodríguez et al., 1999*; *McDaniel and Binder, 2012*). Altogether, these experiments show that inhibition of ethylene biosynthesis upon seed imbibition is not sufficient to block germination and therefore are not consistent with the hypothesis that AMB blocks germination because it blocks ethylene biosynthesis.

AMB also inhibits numerous PLP-dependent enzymes in vitro including aspartate aminotransferase and tryptophan synthase (*Rando, 1974*; *Rando et al., 1976*; *Miles, 1975*). AMB could also inhibit methionyl-transfer RNA synthetase indicating that it could act as a methionine antimetabolite (*Mattoo et al., 1979*).

This could suggest that the effect of AMB to block germination results from its inhibition of auxin or methionine synthesis. The former possibility is unlikely because (1) auxin promotes ABA-dependent repression of seed germination and (2) low auxin levels may facilitate seed germination since ABA-dependent inhibition of radicle elongations involves enhancement of auxin signaling in the radicle elongation zone (*Belin et al., 2009*; *Liu et al., 2013*). We found that *tir1* mutants, deficient in the auxin receptor TIR1, germinate normally, consistent with the report of Liu *et al.* showing that *tir1/afb2* and *tir1/afb3* double mutants, deficient in the auxin receptors TIR1 and AFB1 or TIR1 and AFB3, germinate normally and are less dormant (*Figure 7—figure supplement 5A and B*, *Liu et al., 2013*). Furthermore, WT seeds treated with 500 µM of yucasin, a potent inhibitor of the YUCCA proteins, which are flavin mono-oxygenases oxidizing indole-3–pyruvic acid to indole-3–acetic acid (auxin), did not prevent their germination (*Figure 7—figure supplement 5C*) (*Nishimura et al., 2014*). These data indicate that low auxin signaling or synthesis upon seed imbibition does not prevent seed germination. In addition, *tir1* mutant seed germination responses to AMB-containing WT *PAO1* extracts was similar to that of WT seeds, indicating that AMB-dependent responses in seeds do not require auxin signaling (*Figure 7—figure supplement 5A and B*).

Concerning methionine, previous reports showed that methionine biosynthesis is essential for seed germination (*Gallardo et al., 2002*). We explored whether AMB could block germination by inhibiting methionine synthesis. DL-Propargylgylcine (PAG) is an active site-directed inhibitor of cystathionine γ-synthase, which is necessary for methionine synthesis (*Thompson et al., 1982*). WT seeds treated with 1 mM PAG were unable to germinate, consistent with previous reports (*Gallardo et al., 2002*). As expected and consistent with previous reports, exogenously added methionine in the germination medium fully restored germination in PAG-treated seeds (*Figure 7—figure supplement 6*) (*Gallardo et al., 2002*). However, exogenous methionine did not rescue the germination of *PAO1* treated WT seeds. These observations are not consistent with the hypothesis that AMB prevents germination by limiting methionine synthesis.

Altogether, these results are not supporting the view that AMB exerts its DELLA-dependent seed germination arrest by inhibiting ethylene, auxin or methionine synthesis.

## How AMB invokes a DELLA-dependent germination arrest remains to be understood

Here, we unambiguously identify AMB as the main if not only GRA released by *P. aeruginosa* affecting *Arabidopsis* seed germination. AMB, unlike AVG, represses *Arabidopsis* seed germination and this requires functional DELLA genes.

Our results strongly suggest that AMB does not interfere with GA synthesis. Rather, they suggest that AMB can regulate seed gene expression in both a DELLA-dependent and GA-independent manner (*Figure 3 and 5B*). Our results with RGL2, GAI and RGA suggest that AMB does not interfere with GA-dependent DELLA degradation (*Figures 4B,6A–D* and *Figure 6—figure supplement 1*).

We rather provide genetic evidence that DELLA activity to promote ABA-dependent seed germination arrest is stimulated in presence of AMB (*Piskurewicz et al., 2009*, *2008*). The mechanism through which AMB enhances DELLA activity to promote ABA-dependent responses in seeds remains to be understood. In particular, whether it is the result of a direct interaction between AMB and DELLAs is not known.

Exogenous GA cannot overcome germination repression triggered by *P. aeruginosa* or AMB (*Figures 3*,*4B,6D* and *Figure 6—figure supplement 1*). Thus, genes whose expression is DELLA-dependent and GA-independent in response to *P. aeruginosa* or synthetic AMB might provide clues about the potential mode of action of AMB. In this respect, our transcriptome analysis identified 87 genes whose expression is regulated by DELLA factors in seeds exposed to *P. aeruginosa* extracts (*Figure 3D*, *Supplementary file 2*). Among them, 24 genes had their expression unchanged when GA was included in the medium together with *P. aeruginosa* extracts (*Figure 3D*, *Supplementary file 2*). Interestingly, the expression of 18 of them was not significantly changed in seeds exposed to paclobutrazol (PAC), an inhibitor of GA synthesis, further suggesting these genes are not regulated by GA. In this set of genes, those related to karrikin signaling were overrepresented (*Nelson et al., 2009*; *Waters et al., 2012*). Karrikins are a class of compounds found in the smoke of burning plant material, which are known to promote germination and to break seed dormancy (*Nelson et al., 2009*). Furthermore, seeds lacking the karrikin receptor *KAI2* are dormant (*Waters et al., 2012*). The karrikin signaling genes *KUF1* and *BBX20/STH7* are strongly repressed in WT seeds exposed to *P. aeruginosa* but not in Δ*della* mutant seeds, lacking all DELLA factors (*Supplementary file 2*). Thus, our data suggest that AMB could repress germination by repressing karrikin signaling through the DELLAs. However, karrikins share common signaling components with strigolactones, a class of plant hormones also promoting germination (*De Cuyper et al., 2017*; *Toh et al., 2012*). Strigolactones were recently proposed to regulate GA signaling in rice (*Ito et al., 2017*). SLR1, a rice DELLA factor, was found to interact with the strigolactone receptor DWARF14 fused to its ligand (*Nakamura et al., 2013*). Thus, AMB could also potentially regulate strigolactone signaling in seeds.

## AMB production and release is under the control of *P. aeruginosa's* quorum sensing IQS

Beyond the question of how AMB affects *Arabidopsis* developmental responses, its biological significance in *Pseudomonas* has recently attracted much attention.

Indeed, genetic experiments have shown that IQS, the autoinducer of the recently discovered quorum sensing (QS) subsystem named IQS in *P. aeruginosa*, is controlled by the five-gene operon *ambABCDE* (*Lee et al., 2013a*). Although the IQS receptor remains unknown it also remains to be determined what is the link between *ambABCDE* and IQS synthesis. Indeed, Lee *et al.* first proposed that *ambABCDE* gene products are directly responsible for IQS synthesis and therefore activity of the quorum sensing IQS (*Lee et al., 2013a*). However, the link between *ambABCDE* and IQS synthesis remains to be clarified. Indeed Lee *et al.* showed that *ambABCDE* is also necessary for AMB production and Rojas Murcia *et al.* proposed that *ambABCDE* gene products rather synthesize and export AMB (*Rojas Murcia et al., 2015*; reviewed in *Moradali et al., 2017*; *Lee et al., 2010b*, *2013a*).

In any case, there is no genetic controversy regarding the need of a functional *ambABCDE* operon for (1) IQS production and signaling and (2) AMB synthesis and release by *P. aeruginosa*.

In this study, we further confirm that presence of the *ambABCDE* operon is necessary for AMB production (*Figure 4—figure supplement 5*, *Supplementary file 3*). Furthermore, we show that AMB is also abolished in Δ*lasI*Δ*rhlI P. aeruginosa* mutants lacking the *las* and *rhl* QS subsystems (*Figure 4—figure supplement 5A*, *Supplementary file 3*). This is in agreement with a previous report showing that expression of *amb* operon is strongly downregulated in Δ*lasI*Δ*rhlI* mutants (*Schuster et al., 2003*). The genes of the *ambABCDE* operon were also singled out as quorum-dependent genes in chronic cystic fibrosis patients infected with *P. aeruginosa* (*Chugani et al., 2012*).

Thus, these reports leave little doubt that the *ambABCDE* operon, which is necessary for AMB production, is being intimately linked to the activity of QS in *P. aeruginosa*.

## Potential ecological and evolutionary significance of an AMB-dependent germination arrest

Publicly available genomic sequences show that the *ambABCDE* operon is not only present in the *P. aeruginosa PAO1* strain used in this study but is also present in numerous other *P. aeruginosa* strains (*Figure 4—figure supplement 11*). This raises the question of the biological significance of the AMB-dependent germination arrest involving DELLA factors described here. The biological significance can be divided in two broad categories.

Firstly, AMB could exert its effect fortuitously, that is in an accidental manner that bears no ecological or evolutionary significance. This does not preclude the potential biological interest of the effect of AMB on seeds. Indeed, we provided evidence that the AMB- and DELLA-dependent germination arrest cannot be readily explained by an AMB-dependent inhibition of ethylene, auxin or methionine synthesis. Furthermore, AMB does not appear to prevent GA-dependent DELLA degradation. Given that oxyvinylglycines were reported to inhibit PLP-dependent enzymes and that AMB is a methionine analog, our results could indicate that AMB interferes with an unknown mechanism present in *Arabidopsis* that is linking PLP-dependent enzymes or amino acid metabolism with DELLA factors. Alternatively, AMB could interfere with an unknown and GA-independent mechanism involving DELLA factors to control germination.

Secondly, the effect of AMB on seeds could indeed be ecologically and evolutionary significant. We hereafter discuss this possibility.

*Arabidopsis* produces high seed numbers, which could contradict the need of evolving protective germination arrest responses since one successful germination event is sufficient to maintain the size of the population. However, *Arabidopsis* is not a long distance seed dispersal species and the majority of seeds is expected to fall in the vicinity of the mother plant. Furthermore, *Arabidopsis* seedlings are small and fragile and poor dispersion would increase their chance of being killed at the same time whenever faced by a local threat. It is therefore expected that early development is tightly regulated in *Arabidopsis* to enhance plant survival. Indeed, *Arabidopsis* has evolved elaborate germination arrest control mechanisms that are widely regarded as being protective. These include seed dormancy, believed to prevent germination out of season, and control of seed germination of non-dormant seeds in response to abiotic factors, which is also considered to protect the plant (*Kami et al., 2010*; *Penfield and King, 2009*).

Here, we described laboratory conditions where the AMB-dependent germination arrest protects the plant from the potentially fatal effect of AMB on seedlings (*Figure 7*). This could indicate that this response has evolved as an adaption to counteract damage induced by biotic harmful compounds similarly to what is proposed for abiotic stresses. More generally, it is also consistent with the notion that it could correspond to a protection mechanism against pathogenic bacteria in the environment. Given the link between AMB and QS activity in *P. aeruginosa*, it is tempting to speculate that evolving a germination arrest response to AMB could be doubly advantageous: (1) it could protect the plant from the AMB toxin and (2) it could reveal the presence of the plant pathogen *P. aeruginosa*.

However, presently these considerations remain highly speculative. Indeed, the GRAs released by *P. aeruginosa* reported here are observed after culturing *P. aeruginosa* to high densities that trigger IQS QS activity. Whether bacteria such as *P. aeruginosa* proliferate to such high densities in the rhizosphere is unclear. The number of *P. aeruginosa* cells building up in the environment is subject to controversy. Green *et al.* were able to detect *P. aeruginosa* in 24% of soil samples studied and reported that it multiplied in lettuce and bean under conditions of high temperature and high

humidity (*Green et al., 1974*). On the other hand, Deredjian *et al.* reported that *P. aeruginosa* has low occurrence in agricultural soils. However, they were able to detect them in high amounts in various manures, consistent with previous reports (*Deredjian et al., 2014*). These results could suggest that *P. aeruginosa* could only be found in high densities in the rhizosphere where food is available, including near decaying fruit or animal droppings, together with the proper moisture or temperature conditions. This could limit the ecological significance of controlling seed germination responses to biotic factors.

On the other hand, high densities of bacteria in the rhizosphere may not be obligatory to elicit seed germination responses. The Quorum Sensing is usually invoked to describe situations when high densities of cells trigger coordinated responses after autoinducers reach high concentrations in the environment. However, a given individual bacterium can only detect the autoinducer concentration present in its immediate proximity, which can include the autoinducer molecules that the same bacterium releases. In the rhizosphere, there could be situations limiting autoinducer diffusion, altering autoinducer advection, reducing autoinducer degradation or altering autoinducer spatial distribution, which could lead to high local concentrations of autoinducer even in absence of high densities of bacteria. Thus, a given bacteria cannot distinguish among the various scenarios leading to high autoinducer concentration. These considerations lead to the proposal that autoinducers fulfill a role beyond that of detecting high densities of bacteria (*Redfield, 2002*). They could allow bacteria to sense whether diffusion of molecules in their immediate environment is limited (Diffusion Sensing –DS-). In turn, this would allow a given bacteria to determine whether a given effector would diffuse efficiently or not (*Hense et al., 2007*).

Clearly, testing the model that the AMB-dependent germination arrest fulfills an adaptive function in plants will require future investigations. These include (1) a better understanding of the ecology of *P. aeruginosa* in real field settings, (2) identifying AMB's interacting targets in *Arabidopsis* responsible to convey the AMB-dependent germination arrest and (3) studying the fitness in the field of *Arabidopsis* mutants lacking those targets.

Undoubtedly, the findings reported here offer a very narrow sample of interactions that could take place between seeds and living organisms in the rhizosphere.

## Materials and methods

### Plant material

All seed batches compared in this study were harvested on the same day from plants growth side by side under the same environmental conditions. Seeds of *Arabidopsis thaliana* plants were all from Columbia Col-0 background. The *Arabidopsis* mutants used in this study were *aba1-6* (*Barrero et al., 2005*), *abi5-3* (*Finkelstein and Lynch, 2000*), *abi3-8* (*Nambara et al., 2002*), *tir1-1* (purchased on Nottingham Arabidopsis Stock Centre -NASC- N3798, *Ruegger et al., 1998*), Δ*acs145679* (*acs1-1 acs2-1 acs4-1 acs5-2 acs6-1 acs7-1 acs9-1*, purchased on NASC, N16650, *Tsuchisaka et al., 2009*), *rgl2-13* (*Tyler et al., 2004*), Δ*della* (*rgl2-Sk54 rga-28 gai-t6 rgl1-Sk62 rgl3-3*; *Park et al., 2013*). We generated *rgl2-SK54 rga-28* double mutants and *rgl2-SK54 rga-28 gai-t6* triple mutants for this study. Seeds were surface sterilized and sowed on germination plates as described (*Piskurewicz and Lopez-Molina, 2016*). Germination plates were incubated in growth chambers (22°C, 70% humidity, 100µmol/m$^2$/s, 16 hr/8 h day/light photoperiod).

### *Pseudomonas* material

The *Pseudomonas* strains used in this study are listed in *Supplementary file 5*.

### Preparation of germination plates containing *P. aeruginosa*

Bacteria were cultured with agitation in 5 ml LB medium for 16 hr at 37°C (with the exception of *P. fluorescens*, grown at 30°C). Bacteria density was controlled by OD$_{600}$ (Ultrospec 2000, Pharmabiotec) and stocks were made for each bacteria strain when OD$_{600}$ reached 1.2. A volume of 25 µL of liquid culture containing approx. $7 \times 10^8$ CFU was streaked on a square plate (120 × 120 mm, Huberlab) containing Murashige and Skoog (MS) medium (4.3 g/L), 2-(N-morpholino)ethanesulfonic acid (MES) (0.5 g/L), 0.8% (w/v) Bacto-Agar (Applichem) and 20 mM succinate. Plates were incubated for 3 days in the dark in plant growth chambers (22°C, 70% humidity). The resulting plates

were used for germination tests as described in *Figure 1—figure supplement 1*. All experiments were repeated independently several times with similar results.

## Preparation of germination plates containing bacteria-free extracts

Bacteria were cultured with agitation in a liquid solution (0.215 g/L MS, 20 mM succinate) for 24 hr at 37°C and the saturated culture was centrifuged at 4°C for 30 min (HiCen XL, Herolab) to pellet bacteria. The supernatant was then filtrated (0.22 µm filter) and lyophilized (Freeze-dryer Alpha 2–4 LD plus, Christ) as described in *Figure 2—figure supplement 2*. The resulting bacteria-free lyophilizate (also referred as 'extract' in the main text) was stored at −20°C. To perform germination assays, the lyophilizate was resuspended in water at 40 mg/ml and used at different concentrations as shown and described in the figures and figure legends. All experiments were repeated independently several times with similar results.

## Chemical treatments

Commercially available chemicals used in this study can be find as following: 5-(4-Chlorophenyl)−2,4-dihydro-[1,2,4]-triazole-3-thione (named Yucasin, CAS registry number: 26028-65-9) was ordered from Santa Cruz biotechnology (product sc-233161); DL-Propargylglicine (named PAG, CAS registry number: 64165-64-6), was ordered from Sigma-Aldrich (product P7888), (S)-*trans*-2-Amino-4-(2-aminoethoxy)−3-butenoic acid (named AVG, CAS registry number: 55720-26-8) was ordered from Sigma-Aldrich (product A6685) and AgNO$_3$ (CAS registry number: 7761-88-8) was ordered from Sigma-Aldrich (product S7276).

## Western blots

Seed extracts were prepared as previously described (*Piskurewicz et al., 2008*). Polyclonal anti-RGL2 and anti-ABI5 were as previously described (*Piskurewicz et al., 2008*). Polyclonal anti-GAI was produced as described in *Piskurewicz et al. (2008)*. Anti-RGA antibody was purchased (Agrisera, product AS11 1630, RRID:AB_10749442). A commercial anti-UGPase antibody was used as a loading control (Agrisera, product AS05 086, RRID:AB_1031827).

## RNA extraction and RT-qPCR

Total RNA was extracted as described (*Piskurewicz and Lopez-Molina, 2016*). RNAs were treated with RQ1 RNase-Free DNAse treatment (Promega) and cDNAs were made from 1 ug of RNA using ImProm-II reverse transcriptase (Promega). Amplification was done using GoTaq qPCR Master mix (Promega) and reaction was performed on QuantStudio 5 Real-Time PCR equipment (Thermo Fisher Scientific) according to manufacturer instructions. Relative transcript levels were calculated using the comparative Ct method and normalized to PP2A (AT1G69960) gene transcript levels. qPCR experiment were performed in biological triplicate. Primers used in this study are listed in *Supplementary file 5*.

## RNAseq

Surface-sterilized WT (Col-0) and Δ*della* seeds were sown in germination plates in absence or presence of (PAC 5 µM), *P. aerurinosoa* (*PAO1*) extracts (0.7 mg/mL) or GA (10 µM) and cultured for 20 hr prior to total seed RNA extraction. cDNA libraries from two independent biological replicates were normalized and sequenced using HiSeq4000 (Illumina) with single-end 50 bp reads. Reads were mapped to Col-0 genome (TAIR10) with the TopHat program. Differential gene expression analysis was performed with the Cuffdiff program calculated by pooling the biological replicates. Differentially expressed genes were selected according to their significance in fold-change expression (false discovery rate, FDR < 0.05) and a threshold level of at least two-fold change between samples (log$_2$ ratio $\geq$1 and $\leq$−1). All RNAseq analysis were performed on GALAXY website. For data visualization, clustering analysis were done with Gene Cluster 3.0 using average linkage method and visualized with Java Treeview version 1.1 6r4, were expression levels were color coded as following: red color for overexpressed, black color for unchanged expression and green color for underexpressed genes. All data are publicly available through the GEO database with accession number GSE115272.

## Metabolomic analysis

With the aim to obtain preliminary information about the chemical nature of the GRA, bacterial extracts containing or not containing the GRA were analyzed by comprehensive UHPLC-HRMS[2] metabolite profiling and data were mined by differential untargeted metabolomics (see below Appendix 1, part '1. Metabolomic analysis informations' for details). Data were acquired on extracts from strains releasing (PAO1 WT and ΔpqsA) or not releasing a GRA (ΔambE and ΔlasIΔrhlI). After the appropriate data treatment, all MS signals recorded were gathered as peak lists of individual features (mass and retention time were annotated as follow: m/z @ RT) each strain for subsequent multivariate data analysis to identify biomarkers. After alignment, the resulting peak list (Supplementary file 3) was then mined for differential features using unsupervised PCA (principal component analysis) and supervised statistical analysis approaches OPLS-DA (orthogonal partial least squares discriminant analysis). The PCA already allowed to clearly separate the four different bacteria extracts according to their MS features (Figure 4—figure supplement 3A). An OPLS-DA was carried constructing two 'active' vs. 'non-active' groups namely (PAO and ΔpqsA ('active') vs ΔambE and ΔlasI/rhlI ('non-active') (Figure 4—figure supplement 3B). This analysis afforded a list of biomarkers responsible for the metabolic differences between strains (Supplementary file 4). The annotation of the most significant MS features was done using exact mass information and search against a database of natural products taxonomically restricted to the genus Pseudomonas. The bio-active compound was known to be a polar compound since activity was observed to be present in the $H_2O$ eluted fraction of all GRA-containing extracts when eluted through reversed phase chromatography (Figure 4—figure supplement 6). Taking all this information into account one specific MS feature m/z 327.12 eluting at a retention time of 0.46 min (m/z = 327.12 @ RT 0.46 min) could be highlight when filtering the discriminant loadings of the OPLS-DA analysis for the most polar compounds (Supplementary file 4). This exact mass corresponded to a molecular formula (MF) of $C_{11}H_{22}N_2O_7S$, which did not yield any hit when querying the whole Dictionary of Natural Products (http://dnp.chemnetbase.com/). In order to gain information on this specific feature, a molecular network (MN) was constructed based on MS fragmentation similarities between extracts constituents using the untargeted MS/MS data acquired on PAO, ΔpqsA, ΔambE and ΔlasIΔrhlI mutants and bioactivity data were mapped on this MN. The MN generated allows grouping compounds with structural similarities in clusters (Wang et al., 2016). The MN was searched for the feature m/z = 327.12 @ RT 0.46 min which was found to be related to a cluster of 3 ions (Figure 4—figure supplement 5). As expected the ion m/z 327.12 was only found in the 'active' labelled species. Surprisingly, one of the related ion at m/z 196.06 ($C_6H_{13}NO_4S$) was found in both active and non-active samples but also in the culture media. Using the CSI:FingerID in silico fragmentation platform (https://www.csi-fingerid.uni-jena.de/, Dührkop et al., 2015), it was identified as 2-(N-morpholino)ethanesulfonic acid) or MES, a known constituent of the used culture media. Since MES in known to readily form coordination complexes we focused on the mass difference between ion m/z 327.12 and m/z 196.06, which was found to correspond to a mass difference of 131.1 Da. and a MF of $C_5H_9NO_3$. Searching this MF within reported metabolites of Pseudomonas sp. permitted to annotate this compound, possibly responsible for the GRA, as L-2-amino-4-methoxy-trans-3-butenoic acid (also referred as methoxyvinylglycine or AMB). PCA analysis (Figure 4—figure supplement 4A) followed by an OPLS-DA was also carried between biological replicates of P. aeruginosa extract ΔambE and AMBox (Figure 4—figure supplement 4B). This analysis indicated that m/z 327.12 was found between the most discriminant features (Figure 4—figure supplement 5B, Supplementary file 4). Comparison of the extracted MS ion trace intensities of m/z 327.12 among the mutants indicated that this ion was indeed over-expressed in AMBox (ca. 20 fold between AMBox and PAO1 WT, Figure 4—figure supplement 5B). Additional informations can be found below in the section Appendix 1. All raw data have been deposited under the Massive Dataset ID MSV000082463, available at the following address: ftp://massive.ucsd.edu/MSV000082463.

## Biochemical purification procedures

WT PAO1 and ΔambE extracts were separately fractionated by a reversed phase semi-preparative HPLC (High Performance Liquid Chromatography) into four fractions. The conditions for fractionation were obtained by a gradient chromatographic transfer of the metabolite profiling after optimization at the analytical level (see below Appendix 1, part '2. Bioguided biochemical purification

*analysis'* for details). A GRA was found to be present in the polar fraction F1 from WT *PAO1* extracts only. The NMR analysis of this fraction allowed detecting the presence of characteristic proton signals of the AMB molecule (data not shown). In order to confirm the structure of this compound and assess its biological properties, the crude extract of the strain *AMBox* was fractionated at large scale using RP-MPLC (Reversed Phase Medium Pressure Chromatography, C18) followed by semi-preparative HPLC purification using an amide stationary phase for efficient selectivity (*Figure 4—figure supplement 6B*). The structure of AMB was finally confirmed by extensive 1D and 2D NMR and HRMS analyses (*Appendix 1—figures 1* and *2*). For AMB quantification and NMR analysis, see below Appendix 1 part '3. *AMB quantification*' and part.4 *'Chemical identity of the isolate AMB'* for details.

## Acknowledgements

We are especially grateful to Cornelia Reimmann for generously providing synthetic AMB, various *Pseudomonas aeruginosa* strains and for discussions. We are also indebted to the following colleagues: Stephan Hebb, Miguel Camara and Karl Perron for providing *Pseudomonas* species and *P. aeruginosa* strains. Giltsu Choi provided Δ*della* seeds. Julien De Giorgi and Adrien Sicard provided several brassicaceae seeds. We would like to acknowledge Davide Righi (EPGL) for his great support in the purification of AMB, and Joëlle Houriet and Helena Mannochio Russo (EPGL) for their assistance for the UHPLC analyses. We thank Mylene Docquier and members of the Genomics Platform of the Institute of Genetics and Genomics (iGE3) at the University of Geneva for help with RNAseq experiments. We thank all members of the LLM laboratory and Karl Perron for discussions. This work was supported by grants from the Swiss National Science Foundation and by the State of Geneva (LLM). JLW is grateful to the Swiss National Science Foundation (SNF) for supporting their natural product metabolomics projects (grants nos.: 310030E-164289 and 31003A_163424). NMR characterization of AMB and related metabolites was carried out on the 600 MHz NMR obtained thanks to a SNF R'Equip grant to JLW and LLM (grants no: 316030_164095).

## Additional information

### Funding

| Funder | Grant reference number | Author |
|---|---|---|
| Schweizerischer Nationalfonds zur Förderung der Wissenschaftlichen Forschung | 31003A_152660 | Luis Lopez-Molina |
| Schweizerischer Nationalfonds zur Förderung der Wissenschaftlichen Forschung | 310030E-164289 | Jean-Luc Wolfender |
| Schweizerischer Nationalfonds zur Förderung der Wissenschaftlichen Forschung | 31003A_163424 | Jean-Luc Wolfender |
| Schweizerischer Nationalfonds zur Förderung der Wissenschaftlichen Forschung | 316030_164095 | Jean-Luc Wolfender |

The funders had no role in study design, data collection and interpretation, or the decision to submit the work for publication.

### Author contributions

Hicham Chahtane, Conceptualization, Data curation, Formal analysis, Supervision, Investigation, Methodology, Writing—original draft, Project administration, Writing—review and editing; Thanise Nogueira Füller, Data curation, Methodology, Writing—review and editing; Pierre-Marie Allard, Conceptualization, Resources, Formal analysis, Methodology, Writing—review and editing; Laurence Marcourt, Resources, Methodology, Writing—review and editing; Emerson Ferreira Queiroz, Conceptualization, Formal analysis, Supervision, Methodology, Writing—review and editing; Venkatasalam Shanmugabalaji, Conceptualization, Data curation, Methodology; Jacques Falquet,

Conceptualization, Methodology; Jean-Luc Wolfender, Conceptualization, Resources, Formal analysis, Supervision, Funding acquisition, Writing—review and editing; Luis Lopez-Molina, Conceptualization, Formal analysis, Supervision, Funding acquisition, Validation, Investigation, Methodology, Writing—original draft, Project administration, Writing—review and editing

### Author ORCIDs
Hicham Chahtane (iD) http://orcid.org/0000-0002-0976-1186
Pierre-Marie Allard (iD) http://orcid.org/0000-0003-3389-2191
Emerson Ferreira Queiroz (iD) http://orcid.org/0000-0001-9567-1664
Jean-Luc Wolfender (iD) http://orcid.org/0000-0002-0125-952X
Luis Lopez-Molina (iD) http://orcid.org/0000-0003-0463-1187

### Decision letter and Author response
Decision letter https://doi.org/10.7554/eLife.37082.059
Author response https://doi.org/10.7554/eLife.37082.060

## Additional files
### Supplementary files
• Supplementary file 1. List of differentially expressed genes in WT seeds in response to PAC.
DOI: https://doi.org/10.7554/eLife.37082.040

• Supplementary file 2. List of differentially expressed genes in WT seeds in response to *PAO1* extract.
DOI: https://doi.org/10.7554/eLife.37082.041

• Supplementary file 3. Peaklist of features detected by MS/MS in the different *P. aeruginosa* extracts.
DOI: https://doi.org/10.7554/eLife.37082.042

• Supplementary file 4. Peaklist of most discriminant features determined by OPLSDA analysis in active versus inactive extracts.
DOI: https://doi.org/10.7554/eLife.37082.043

• Supplementary file 5. Bacterial strains and oligonucleotides.
DOI: https://doi.org/10.7554/eLife.37082.044

• Supplementary file 6.
DOI: https://doi.org/10.7554/eLife.37082.045

• Supplementary file 7.
DOI: https://doi.org/10.7554/eLife.37082.046

• Transparent reporting form
DOI: https://doi.org/10.7554/eLife.37082.047

### Data availability
Sequencing data have been deposited publicly through the GEO database under accession number GSE115272. Metabolomic data have been deposited under the Massive Dataset ID MSV000082463, available at the following address: ftp://massive.ucsd.edu/MSV000082463 The generated molecular network is accessible at the following address: https://gnps.ucsd.edu/ProteoSAFe/status.jsp?task=98375138ca634aa2827d693d3235ebe5 Source data files have been provided for Figures 1, 2, 3, 4 and 7.

The following datasets were generated:

| Author(s) | Year | Dataset title | Dataset URL | Database, license, and accessibility information |
|---|---|---|---|---|
| Chahtane H | 2018 | Transcriptomic analysis | http://www.ncbi.nlm.nih.gov/geo/query/acc.cgi?acc=GSE115272 | Publicly available at the NCBI Gene Expression Omnibus (accession no: GSE115272). |

| Allard PM | 2018 | Metabolomic analysis | ftp://massive.ucsd.edu/MSV000082463 | Publicly available for download at the Massive Dataset website (ID: MSV000082463) |
| Allard PM | 2018 | Molecular network | https://gnps.ucsd.edu/ProteoSAFe/status.jsp?task=98375138ca634aa2827-d693d3235ebe5 | Publicly available at the GNPS: Global Natural Products Social Molecular Networking website |

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

# Appendix 1

DOI: https://doi.org/10.7554/eLife.37082.048

## Metabolomic analysis informations

### Extraction procedure

Bacteria-free extracts were solubilized in water (4 mg/ml) for mass spectrometry analysis.

### Data dependent UHPLC-HRMS$^2$ analysis

Chromatographic separation was performed on an Acquity UHPLC system (Waters, Milford, MA, USA) interfaced to a Q-Exactive Plus mass spectrometer (Thermo Scientific, Bremen, Germany), using a heated electrospray ionization (HESI-II) source. The LC conditions were as follows: column: Waters BEH C18 100 × 2.1 mm, 1.7 μm; mobile phase: (A) water with 0.1% formic acid; (B) acetonitrile with 0.1% formic acid; flow rate: 600 μL.min-1; injection volume: 1 μL; gradient: linear gradient of 5–100% B over 8 min and isocratic at 100% B for 3 min. In positive ion mode, diisooctyl phthalate $C_{24}H_{38}O_4$ [M + H]$^+$ ion ($m/z$ 391.28429) was used as internal lock mass. The optimized HESI-II parameters were the following: source voltage: 3.5 kV (pos), sheath gas flow rate (N2): 48 units; auxiliary gas flow rate: 11 units; spare gas flow rate: 2.0; capillary temperature: 300°C (pos), S-Lens RF Level: 55. The mass analyzer was calibrated using a mixture of caffeine, methionine-arginine-phenylalanine-alanine-acetate (MRFA), sodium dodecyl sulfate, sodium taurocholate and Ultramark 1621 in an acetonitrile/methanol/water solution containing 1% formic acid by direct injection. The data-dependent MS/MS events were performed on the four most intense ions detected in full scan MS (Top4 experiment). The MS/MS isolation window width was 2 Da, and the normalized collision energy (NCE) was set to 35 units. In data-dependent MS/MS experiments, full scans were acquired at a resolution of 35 000 FWHM (at $m/z$ 200) and MS/MS scans at 17 500 FWHM both with a maximum injection time of 50 ms. After being acquired in the MS/MS scans, parent ions were placed in a dynamic exclusion list for 3.0 s.

### Molecular Network Analysis

The MS$^2$ data were converted from the. RAW (Thermo) or. d (Agilent) standard data-format to. mzXML format using the MSConvert software, part of the ProteoWizard package. *Chambers et al., 2012* The molecular network was created using the online workflow at GNPS (http://gnps.ucsd.edu). The data was then clustered with MS-Cluster with a parent mass tolerance of 0.08 Da and a MS/MS fragment ion tolerance of 0.05 Da to create consensus spectra. Further, consensus spectra that contained less than two spectra were discarded. A network was then created where edges were filtered to have a cosine score above 0.65 and more than six matched peaks. Further edges between two nodes were kept in the network if and only if each of the nodes appeared in each other's respective top 10 most similar nodes. The spectra in the network were then searched against GNPS' spectral libraries. All matches kept between network spectra and library spectra were required to have a score above 0.6 and at least six matched peaks.

The generated molecular network is accessible at the following address: https://gnps.ucsd.edu/ProteoSAFe/status.jsp?task=98375138ca634aa2827d693d3235ebe5

### MS data treatment and multivariate data analysis

MzMine data treatment parameters are available as a. txt parameter file for both data sets: (*PAO1* and *ΔpqsA*) vs. (*ΔambE* and *ΔlasIΔrhlI*) analysis (*Supplementary file 6*) and *ΔambE* and *AMBox* analysis, (*Supplementary file 7*).

For the (*PAO1* and *ΔpqsA*) vs. (*ΔambE* and *Δlasl ΔrhlI*) analysis, this data treatment was led on two biological replicates and led to a peaklist of 1382 individual features (m/z@RT) (**Supplementary file 3**). For the *ΔambE* vs. *AMBox* analysis, this data treatment was led on three biological replicates and led to a peaklist of 1049 individual features (m/z@RT) (**Supplementary file 3**). PCA and supervised multivariate data analysis was done using the ropls R package available at (https://doi.org/10.18129/B9.bioc.ropls). Mean-centering and pareto scaling were applied. Significance of the respective models was depicted in the **Figure 4—figure supplement 3** and **Figure 4—figure supplement 4**, respectively.

## Bioguided biochemical purification analysis

### HPLC-PDA-ELSD analysis

HPLC-PDA-ELSD analyses for the profiling the *Pseudomonas* extracts prior to isolation were conducted on a HP 1260 system equipped with a photodiode array detector (Agilent Technologies, Santa Clara, CA, USA) connected to an ELSD Sedex 85 (Sedere, Oliver, France). The HPLC conditions were as follows: Interchim PF10 C18 column (250 × 4.6 mm i.d., 10 µm, Montluçon, France); solvent system MeOH (B) and $H_2O$ (A). Flow rate 1 mL/min; injection volume 20 µL; sample concentration 10 mg/mL in the mobile phase. The UV absorbance was measured at 210, 254, 280 and 366 nm and the UV-PDA spectra were recorded between 190 and 600 nm (step 2 nm). The ELSD detection parameters were: pressure 3.5 bar, 40°C, gain 8.

### Semi prep HPLC-UV fractionation of the PAO1 extract

The fractionation of the crude ethanol extract using the semi-preparative HPLC-UV (High Performance Liquid Chromatography hyphenated to Ultra-Violet detector) was performed with an X-Bridge RP C18 column (5 µm, 250 × 21.2 i.d., mm; Waters, Milford MA, USA). The flow rate was set at 17 mL/min and the injection volume was of 500 µL (30 mg of the crude extract). The solvent system used was (A) $H_2O$ and (B) MeOH. The separation was performed using the follow conditions: 3% of B between 0 and 15 min, follow by gradient step of 3% of B to 100% of B in 45 min. The column was washed for 10 min with 100% B. 75 fractions of 12 ml were collected (see separation in **Figure 4—figure supplement 6A**). Eight consecutive injections were performed (240 mg of the crude extract in total). After collection, each fraction was evaporated to dryness using a SpeedVac (HT-4X Genevac, Stone Ridge, NY, USA). In order to obtain a rough fractionation and identify the zone of the chromatogram containing bioactive constituents, the fractions were combined in four main fractions (F1-F4) with the following yield F1 (142.2 mg), F2 (9.88 mg), F3 (6.76 mg) and F4 (18.91 mg). The GRA activity of the four fractions was evaluated in the germination assay. Fraction F1 presented a strong GRA and proton NMR signals corresponding to AMB.

### Isolation of the germination inhibitor principle from the AMBox extract

The lyophilized supernatant (4.2 g) of the *AMBox* strain was fractionated using medium pressure liquid chromatography (MPLC-UV) with a Zeoprep C18 column (40–63 m, 460 × 49 mm i.d.; Zeochem, Uetikon am See, Switzerland). These conditions were first optimized on an analytical HPLC column (250 × 4.6 mm i.d., 15–25 µm, Zeochem, Uetikon am See, Switzerland) packed with the same stationary phase and then chromatographic conditions were geometrically transferred to the preparative scale (**Challal et al., 2015**). The extract was introduced in the MPLC column by dry injection. For this, 4.2 g of the extract were mixed with 20 g of Zeoprep C18 stationary phase. The mixture was conditioned in a dry load cell (11.5 × 2.7 cm i.d.). The dry load cell was connected between the pump and the MPLC column. The flow rate was 30 mL/min, and the UV absorbance was detected at 210, 254 and 366 nm. The solvent system used was (A) $H_2O$ and (B) MeOH. The separation was performed using the following conditions: 3% of B between 0 and 69 min, 3% to 40% of B in 162 min, 40% of B during 63 min, and a final step of 40% to 100% of B in 126 min. 46 fractions of 250 ml were

collected. After collection, each fraction was evaporated to dryness using a Syncore Analyst (Buchi, Flawil, Switzerland). The fractions were controlled by HPLC-UV Fraction 3 (511.3 mg) was chosen to be purified since it provided the same chemical profile as the active previously detected at the bio-guided purification phase described above. The final purification step was performed by semi-preparative HPLC using an X-Bridge BEH Prep OBD Amide column (5 μm, 250 × 19 mm i.d.; Waters, Milford MA, USA) using (A) Acetonitrile/water 95:5 with 0.4% FA and (B) $H_2O$ with 0.2% FA and 100 mM ammonium formate. The flow rate was 11 mL/min, and the UV absorbance was detected by UV scan (210–366 nm). All fractions were submitted to UHPLC-ToF analysis and proton NMR. Using this approach AMB (2.5 mg) was purified from F37 (*Figure 4—figure supplement 6B*).

## Fraction control by UHPLC-ToF analysis

Aliquots (0.1 ml) of each semi-prep HPLC fractions were diluted with 400 μl and analyzed by UHPLC-ToF (Ultra High Performance Chromatography hyphenated to Time of Flight Mass Spectrometer) using the followed conditions: ESI conditions were as follows: capillary voltage 2400 V, cone voltage 40 V, MCP detector voltage 2500 V, source temperature 120°C, desolvation temperature 300°C, cone gas flow 20 L/h, and desolvation gas flow 700 L/h. Detection was performed both in negative and positive ion mode with a *m/z* range of 100–1300 Da and a scan time of 0.5 s in the W-mode. The MS was calibrated using sodium formate, and leucine encephalin (Sigma-Aldrich, Steinheim, Germany) was used as an internal reference at 2 μg/mL and infused through a Lock Spray probe at a flow rate of 10 μL/min with the help of a second LC pump. The separation was performed on an Acquity BEH C18 UPLC column (1.7 μm, 50 × 1.0 mm i.d.; Waters, Milford, MA, USA) using a linear gradient (solvent system: A) 0.1% formic acid–$H_2O$, B) 0.1% formic acid–acetonitrile; gradient: 5–95% B in 4.8 min, flow rate 0.3 ml/min). The temperature was set to 40°C. The injected volume was kept constant (2 μL).

# AMB quantification

## Standard Preparation and Calibration Curve

Synthetic standard of AMB (L-2-amino-4-methoxy-*trans*-3-butenoic acid) was dissolved in $H_2O$ solution at 10 mM, and used as standard stock solution for generating calibration curves. The stock solutions were diluted in $H_2O$ to afford 10, 20, 30, 40, 50, 60, 70, 80, 90 and 100 μM solutions of AMB. These 10 standard solutions were injected in triplicate to generate a ten point calibration curve. Standard curve was linear with $R^2$=0.94625 and the following function y = 19175 x - 184455. Peak areas of the target compound in experimental samples were within the linear range of the curve except those of the *AMBox* extract samples, which were diluted 10 fold to fit within the linear range.

## MS conditions for AMB quantification

Chromatographic separation was performed on an Acquity UHPLC system (Waters, Milford, MA, USA) interfaced to a Q-Exactive Focus mass spectrometer (Thermo Scientific, Bremen, Germany), using a heated electrospray ionization (HESI-II) source. The LC conditions were as follows: column: Waters BEH Amide 50 × 2.1 mm, 1.7 μm; mobile phase: (A) water with 20 mM ammonium formate, pH 3; (B) acetonitrile with 0.1% formic acid; flow rate: 400 μL.min-1; injection volume: 1 μL; gradient: isocratic at 100% B for 0.2 min followed by a linear gradient of 100–70% B over 4 min and isocratic at 70% B for 0.6 min. In positive ion mode, diisooctyl phthalate $C_{24}H_{38}O_4$ $[M + H]^+$ ion (*m/z* 391.28429) was used as internal lock mass. The optimized HESI-II parameters were the following: source voltage: 3.5 kV (pos), sheath gas flow rate (N2): 48 units; auxiliary gas flow rate: 11 units; spare gas flow rate: 2.0; capillary temperature: 256.2°C (pos), S-Lens RF Level: 45. The mass analyzer was calibrated using a mixture of caffeine, methionine-arginine-phenylalanine-alanine-acetate (MRFA), sodium dodecyl sulfate, sodium taurocholate and Ultramark 1621 in an acetonitrile/methanol/water

solution containing 1% formic acid by direct injection. The data-dependent MS/MS events were performed on the three most intense ions detected in full scan MS (Top3 experiment). The MS/MS isolation window width was 1 Da, and the normalized collision energy (NCE) was set to 15, 30 and 45 units. In data-dependent MS/MS experiments, full scans were acquired at a resolution of 35 000 FWHM (at *m/z* 200) and MS/MS scans at 17 500 FWHM both with an automatic maximum injection time. After being acquired in the MS/MS scans, parent ions were placed in a dynamic exclusion list for 2.0 s.

## Chemical identity of the isolated AMB

The chemical identification of the isolated AMB was performed by classical Nuclear Magnetic Resonance (NMR) and high resolution mass spectrometry (HRMS) analysis. For this, a series of NMR experiences were recorded on a Bruker Avance III HD 600 MHz NMR spectrometer equipped with a QCI 5 mm Cryoprobe and a SampleJet automated sample changer (Bruker BioSpin, Rheinstetten, Germany). Chemical shifts are reported in parts per million ($\delta$) using the TSP-$d_4$ (sodium trimethlysilyl propionate-d4) signal at $\delta_H$ 0.0; $\delta_C$ 0.0 as reference for $^1$H and $^{13}$C NMR, respectively, and coupling constants ($J$) are reported in Hz. Complete assignments were obtained based on 2D-NMR experiments: Correlation Spectroscopy (COSY), Heteronuclear Single Quantum Correlation (HSQC) and Heteronuclear Multiple Bond Correlation (HMBC) (*Appendix 1—figure 1*). The HRMS of F37 presented a molecular ion *m/z* 130.0498 [M-H]$^-$ corresponding to the molecular formulae $C_5H_9NO_3$. The $^1$H and HSQC NMR spectra of F37 showed two olefinic protons at $\delta_H$ 6.69 (d, $J$ = 12.6 Hz, H-4) / $\delta_C$ 154.5 and 4.92 (dd, $J$ = 12.6, 9.4 Hz, H-3) / $\delta_C$ 103.2, a methine at $\delta_H$ 3.96 (d, $J$ = 9.4 Hz, H-2) / $\delta_C$ 57.5 and a methoxy group at $\delta_H$ 3.60 (s, OMe) / $\delta_C$ 59.1. The COSY correlations from H-2 to H-3 and H-3 to H-4 and the HMBC correlations from H-3 to the methoxy at $\delta_C$ 59.0 and from H-2 to the carbonyl acid at $\delta_C$ 180.7. All these data allowed identifying F37 as L-2-amino-4-methoxy-*trans*-3-butenoic acid (AMB). In order to confirm this hypothesis the NMR spectrum of a synthetic AMB was recorded and compared to the isolated compound (*Appendix 1—figure 2*). Both compound presented the same signals confirming the identity of the isolated compound as AMB.

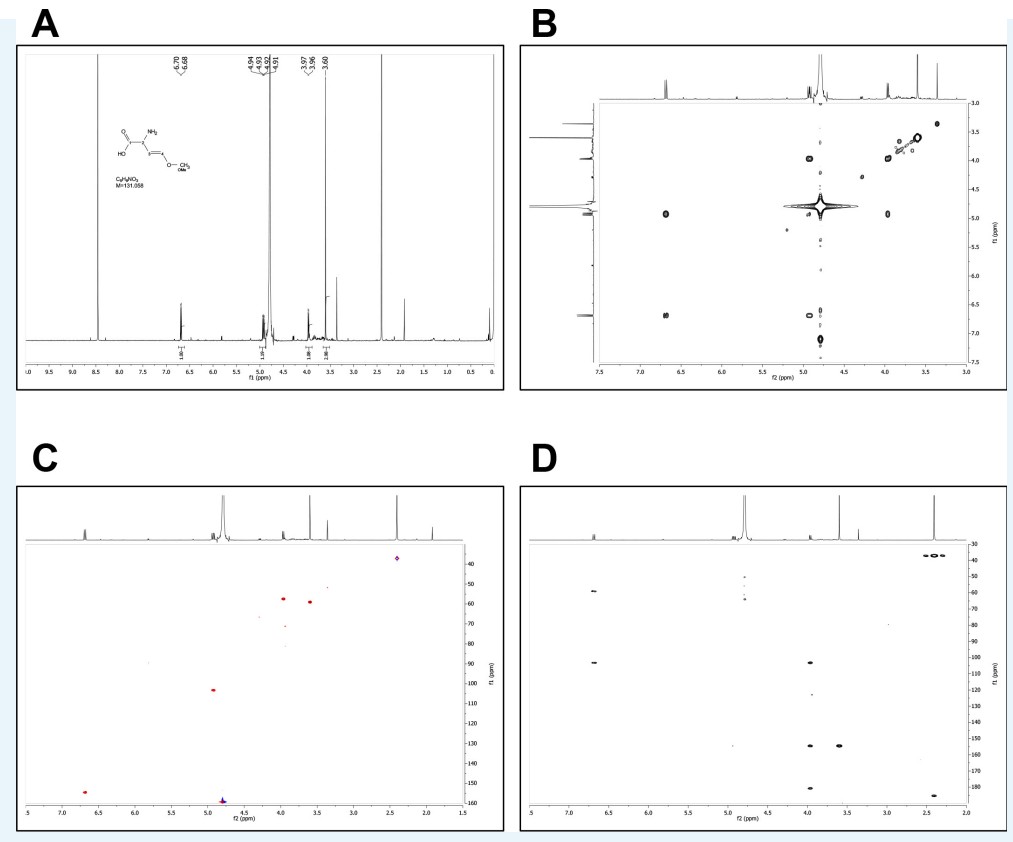

**Appendix 1—figure 1.** Characterization of the fraction F37 by NMR. (**A**) $^1$H NMR spectrum of F37 at 600 MHz in D$_2$O. (**B**) COSY NMR spectrum of F37 in D$_2$O. (**C**) Edited HSQC NMR spectrum of F37 in D$_2$O. (**D**) HMBC NMR spectrum of F37 in D$_2$O.

DOI: https://doi.org/10.7554/eLife.37082.049

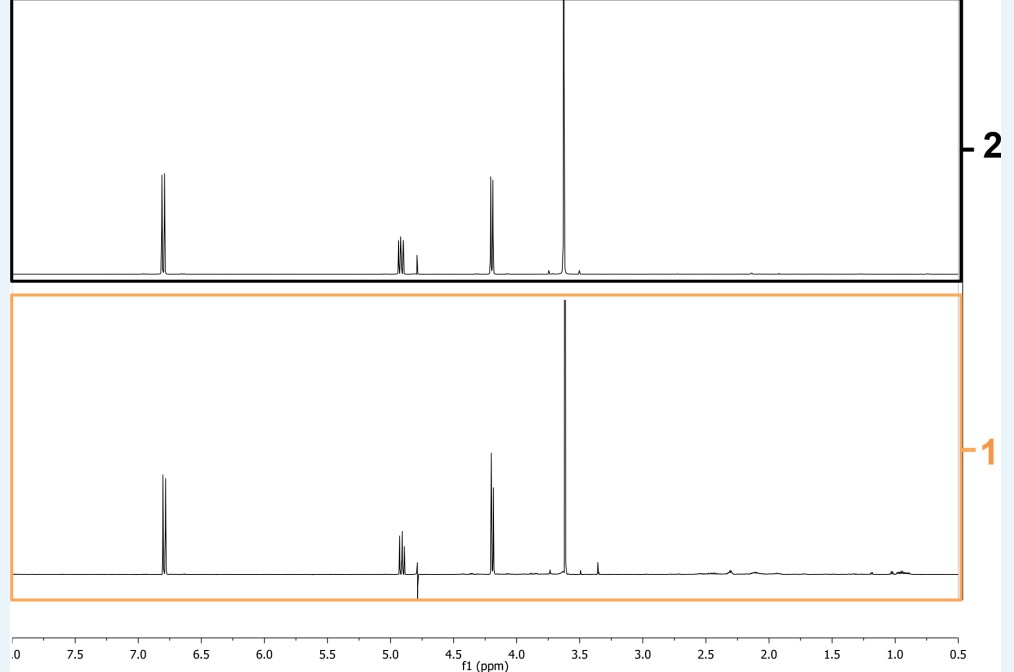

**Appendix 1—figure 2.** $^1$H NMR spectra of F37 and synthetic AMB in D$_2$O. The $^1$H NMR spectra of AMB localized in F37 (1) is similar of that of synthetic AMB, demonstrating that F37 contains AMB.

DOI: https://doi.org/10.7554/eLife.37082.050

Chemical description of AMB (F37): [1]H NMR ($D_2O$, 600 MHz) δ 3.60 (3H, s, OMe), 3.96 (1H, d, $J$ = 9.4 Hz, H-2), 4.92 (1H, dd, $J$ = 12.6, 9.4 Hz, H-3), 6.69 (1H, d, $J$ = 12.6 Hz, H-4). [13]C NMR ($D_2O$, 151 MHz) δ 57.5 (C-2), 59.1 (OMe), 103.2 (C-3), 154.5 (C-4), 180.7 (C-1). HREIMS m/z 130.0498 (calcd for $C_5H_9NO_3$, 130.0510, Δ = −8.7 ppm).

