## [Decision Letter]

Thank you for submitting your article "The plant pathogen *Pseudomonas aeruginosa* triggers a protective DELLA-dependent seed germination arrest in *Arabidopsis*" for consideration by *eLife*. Your article has been reviewed by three peer reviewers, one of whom is a member of our Board of Reviewing Editors, and the evaluation has been overseen by Christian Hardtke as the Senior Editor.

The reviewers have discussed the reviews with one another and the Reviewing Editor has drafted this decision to help you prepare a revised submission.

Summary:

This manuscript works to identify a bacterial metabolite that inhibits *Arabidopsis* germination and proceeds to begin identifying potential mechanisms.

Essential revisions:

1) Additional evidence on the directness of the AMB/GA link is still unproven considering the previously published AMB/AVG links to ethylene and auxin. This could be provided by either evidence of the directness of the relationship, i.e. does AMB inhibit GA-Dependent Della degradation. Or alternatively assessment of the roles of ethylene or auxin in linking AMB to GA signalling.

2) How does this mechanism relate to other oxyvinylglycine metabolites that are widespread? Are other AMB/AVG's functioning by a similar pathway or alternative is AMB altering seedling germination in a range of plants where it is in a known pathogen? The limit on one bacteria to one plant inhibits the interpretation of the ecology and evolution.

3) Clean up the citations on cloning the AMB pathway.

4) Clarify the potential for alternative ecological and evolutionary models that are equally potential.

Reviewer #1:

In this manuscript, the authors describe the identification of an anti-germination metabolite from *P. aeruginosa*. The metabolite, AMB, affects germination via the Della system. The link of compound to germination appears fairly tight but there are alternative hypothesis that could equally explain the potential ecology/evolution of this connection that are not discussed by the authors. Further, it is not clear what this system may mean in the natural environment. There are additional datastreams that the authors could utilize to support their arguments. Presently, the ecology and evolutionary arguments are not supported.

A central concern that I have about the manuscript is the assumption that AMB inhibiting germination is a "protective" event. This arises from several issues. First, does *Arabidopsis* really follow a fragile seedling ecology? A single plant will produce thousands of seeds and only a single seed needs to proceed to maturity to maintain the population size. So, it is not really clear that the "mammalian" protects your progeny eco-evo thoughts really apply to this situation. This eco-evo thought really only applies when maternal reproduction is limiting on an organism's population growth potential. The authors should better discuss the ecology and evolution of seedling defense under a more accurate representation of the system.

Secondly, how frequent does *P. aeruginosa* build up in the environment to the level needed to produce nearly 100 μm concentration surrounding an *Arabidopsis* seed? Especially as this is controlled by quorum sensing. The authors could survey published microbiome work to assess how often *P. aeruginosa* is seen at high levels in the rhizosphere. The simple presence of an observed phenomena does not prove that there are hidden massive populations of this bacteria, especially as there are alternative explanations for the data as described below.

Part of the solution to this concern is that AMB is a part of a larger family of oxyvinyl glycines. These compounds are known from a number of plant associated bacteria and they are thought to inhibit methionine metabolism. This raises two issues, the first is that the authors could support their argument about protective effect by doing a broader survey of plant associated bacteria for oxyvinyl glycine production and test one or two additional mimetics. This would allow them to expand their arguments vastly beyond *P. aeruginosa* and make a more ecological argument. Further, it would mean that it is the level of total oxyvinyl glycines being produced in an environment and not those solely from *P. aeruginosa*.

However, equally, the fact that these compounds are likely methionine synthesis inhibitors. This will conversely affect ethylene production which agrees with the pictures in Figure 6A with altered apical hook formation. This means that the authors need to consider the potential that the AMB is really stimulating the known Ethylene to Della links via an alteration in plant amino acid metabolism. The presented data shows that AMB works via Della but not that it interacts with Della. This is key to the story as if this is the mechanism, then it changes the interpretation from the Plant evolving a connection to the bacteria simply tapping into a mechanism that already existed in the plant.

The authors should comment on if their Psedomonad species individuals in Figure 1 have an oxyvinyl glycine operon. Optimally, they would query across the available genomes to see if other members of these species have an operon.

RNAseq – It is not clear if there is any replication in the RNAseq analysis. The level of independent biological replication needs to be provided to the reader to allow them to evaluate the data. The same is equally true of the RT-qPCR analysis as it is not clear what is an independent biological versus technical replicate. The two need to be handled very differently.

Subsection “How AMB interferes with GA signaling remains to be understood” – Please remember that *Arabidopsis* does not have fire ecology and no need for smoke compounds to affect germination. As such, the Karrikin to seed relationship in *Arabidopsis* has no link to smoke.

Reviewer #2:

Chahtane et al., report that AMB secreted by *P. aeruginosa* blocks DELLA-dependent GA signaling, which in turn inhibits *Arabidopsis* germination. The authors show AMB synthesis in *P. aeruginosa* is controlled by the quorum sensing system. The authors also show that the germination arrest is a protective mechanism for plants to avoid severe seedling damage.

This work investigated seed germination under biotic stress conditions. They tested the germinability of *Arabidopsis* together with four Psuedomonas species and *E. coli* and found only *P. aeruginosa* grown close to seeds inhibit seed germination.

Both bacterial genetic approach and metabolomic approach identified AMB, an oxyvinylglycine, is the primary chemical to inhibits seed germination. It has been shown that AMB released by *P. aeruginosa* inhibits growth of Erwinia, and some oxyvinylglycines inhibit seed germination.

This work includes interesting findings. Oxyvinylglycines released by Psuedomonas have been shown to alter the growth of neighbors, including seeds. It seems that Psuedomonas utilizes AMB to control other organisms surrounding itself. These previous publications provide a credibility of this work to be biological relevant.

The authors discuss AMB acts through the DELLA-dependent pathway (and also GA-independent other pathways). This work will be more attractive if the authors investigates how AMB inhibits GA-mediated DELLA accumulation. The biochemical analysis to investigate if DELLA accumulation observed is reduced degradation or enhanced protein production would be worth to be included, even though the authors are not able to specify the target(s) of AMB.

Reviewer #3:

This study investigates one mechanism by which *Arabidopsis* seeds eavesdrop on the presence of bacterial pathogens in the soil. Two levels of analyses are considered: (i) the chemical nature of the germination-inhibiting small molecule (AMB) (as well as the dependency of AMB synthesis on the 5-gene operon ambABCDE) produced by *Pseudomonas* aeruginosa (PAO), a plant pathogen whose presence in close proximity to *Arabidopsis* seeds blocks their germination; and (ii) the mechanism by which AMB modulates GA signaling and thereby germination. Specifically, authors provide clear evidence that AMB is the central small molecule responsible for the strong germination repressive activity (GRA) of PAO on *Arabidopsis* seeds. Furthermore, using Western blot analyses for RGL2 (a DELLA factor with central function during germination) and ABI5 (an RGL2-controlled ABA signaling transcription factor known to repress germination) as well as germination bioassays with related mutants, authors demonstrate that the GRA of PAO is linked to a prolonged RGL2 stability that promotes the accumulation of ABI5. Overall the study is an interesting merger between germination bioassays, the use of signaling mutants and metabolomics of bacterial natural products to address seeds' germination arrestment in response to perception of soluble bacterial cues. Nonetheless, a weakness of this study is that it does not provide a clear mechanism for how AMB interferes with GA signaling and that it is completely based on assumption that "RGL2 can be used a representative DELLA factor" in the context of this germination arrestment response, while germination of rgl2 as shown in Figure 2A is still strongly compromised by the presence of PAO.

The supplemental material dealing with the differential metabolomics studies of bacteria-free extracts, isolation of the GRA-causing molecules, chemical identification and molecular networking analysis of AMB clearly supports the identification of AMB as the key metabolite responsible for the GRA of PAO on *Arabidopsis* seeds. However, while interesting in order to establish links between seed germination arrestment and bacterial quorum sensing, the part dealing with the dependency of AMB synthesis on the 5-gene operon ambABCDE is essentially confirmatory to previous published work referenced in the text as Lee et al., 2010b and 2013b. In the current version of the text, this did not readily appear clear to me. At least, authors should more rigorously report on previous studies dealing with the operon-dependent synthesis of AMB.

A striking result of this study is that the GRA of PAO is almost fully abolished for mutant seeds lacking all five DELLA factors. This is a key result as it establishes a link between the GRA of PAO (and AMB) and GA signaling. However, in the same figure panel, it is also clear that germination success only mildly increased (compared to that of WT seeds) for rgl2 seeds growing in proximity of PAO. Therefore, regulation of RGL2 levels is not sufficient to explain germination arrestment by PAO. According to the mechanism proposed by this study to explain the GRA of PAO, that AMB-triggered prolonged RGL2 stability promotes the accumulation of the repressing factor ABI5, one would have expected a much stronger germination rate for PAO-challenged rgl2 seeds. In my view, this challenges the use of RGL2 as unique representative DELLA factor throughout the study. Hence, it would have been interesting to monitor, in the context of the different germination/treatment bioassays, levels of other DELLA proteins. Along with this, are there any other DELLA genes present in the list of 130 PAO1-regulated genes?

Interestingly, it is known from previous studies that a prominent plant target of AMB is the ethylene biosynthesis enzyme ACC synthase. Ethylene is required in seeds of many species for dormancy release or germination under optimal or adverse conditions. In particular, ethylene is known to inhibit ABA signaling in seeds to allow germination. The hypothesis that a direct inhibitory activity of AMB over ethylene synthesis could be responsible for the deregulation in ABA signaling observed in non-germinating seeds is not discussed in this study. Did the authors identify signatures of ethylene signaling deregulation in the PAO transcriptome data-set?

[Editors' note: further revisions were requested prior to acceptance, as described below.]

Thank you for submitting your article "The plant pathogen *Pseudomonas aeruginosa* triggers a protective DELLA-dependent seed germination arrest in *Arabidopsis*" for consideration by *eLife*. Your article has been reviewed by three peer reviewers, one of whom is a member of our Board of Reviewing Editors, and the evaluation has been overseen by Christian Hardtke as the Senior Editor. The following individual involved in review of your submission has agreed to reveal his identity: Eiji Nambara (Reviewer #2).

The reviewers have discussed the reviews with one another and the Reviewing Editor has drafted this decision to help you prepare a revised submission. The revision will not go out to external review as it is largely clarifications and caveating.

Summary:

This work begins to study how cross-kingdom signals are sent from bacteria to plants to alter seed germination.

Essential revisions:

1) There is a technical concern in that the RNAseq is still not fully described.

2) The writing about DELLA/AMB needs to be clarified at each instance to show that this is solely a genetic link at this point and has not yet been proven to be a mechanistic link.

3) The ecological and evolutionary ramifications need further qualifications as per the reviewers’ comments.

Reviewer #1:

I still have a technical concern. For the RNAseq, what FDR threshold and other settings were utilized within Cuffdiff to claim significance of gene expression? Also, did the authors utilize significance and fold-change to call differential transcript abundance or only fold-change? It is not clear from the methods.

I think the authors misunderstood my question about the conservation of the AMB cluster within *Psuedomonas aeruginosa*. I was asking both about the distribution of this cluster in other species but also is this cluster conserved across other *P. aeruginosa* or is it specific to this single genotype of *P. aeruginosa*. There are a large number of developing studies on whole genome of wild collected Pseudomonads that could be queried. Right now, the manuscript assumes that this mechanism is widespread throughout all *P. aeruginosa* but it is not clear if this is true. Most pathogens have natural variation in their potential mechanisms and the frequency of the alleles is important to understand the importance. Right now, the authors have one single genotype of one pathogen with this mechanism.

This is key because the seed germination protection models that are discussed in Kami et al., and Penfield and King are linked to abiotic factors that are wide-spread and universally indicative of a specific environment, light, drought, smoke, etc. A highly limited metabolite that may not accumulate to high levels in the wild environment has a much different evolutionary implication. This gets at the question of if this is truly the plants intended response or something that is serendipitous "protection". The authors argue that because it happens, it must be biologically and environmentally and evolutionarily structured to have occurred. The simplest comparison that I can create is that Aspirin can alleviate pain in humans. Yet there is absolutely no reason to presume that willow trees were evolutionarily selected to create methyl salicylic acid (aspirin) just for humans benefit. Nor that humans evolved to eat willow bark to alleviate pain. This specific interaction is likely evolutionary serendipity. This is simply luck that methyl salicylic acid interacts with a single protein conserved amongst animals. Thus, the compound even serendipitously functions in other animals.

I appreciate that the authors have included a section that is meant to present alternative ideas, yet this section is solely built around arguing why their result must be true without ever mentioning the potential for this to be a serendipitous event with no field relevance. A final example of serendipity is the Karrikin receptor. There is no ecological/evolutionary role of smoke perception in *Arabidopsis* and as such, the ability to perceive these signals from smoke is serendipitous. The authors never simply state that the key experiment to prove their model is to find the gene mediating this interaction and to then test if it alters fitness in the field with regards to seedling survival. The strength of this manuscript is the novel interaction and how it is occurring. Yet the authors insist on making the strength of the Introduction and Discussion section being about the protection function.

I'm not sure that extracting saturated rich media grown with *P. aeruginosa* is a reflection on soil densities. I was simply asking the authors to estimate the production potential of a *P. aeruginosa* cell and then query the literature for what range of *P. aeruginosa* cell densities have been found in soil. The plate assay would require a patch of soil to basically be all *P. aeruginosa*. Occurrence is not the same as cell density. Rather than have two paragraphs in the Discussion section arguing that the compound must build up to the needed level, why not simply say that it is critical to understand the cellular ecology of the pathogen in real field settings to test if this mechanism has any potential?

The authors state that "We think it is reasonable to assume that *P. aeruginosa* will be found in high densities in the rhizosphere particularly if food is available together with the proper moisture or temperature conditions. Decaying fruit/animal droppings". Which is possible but doesn't fit into the ecology of the *A. thaliana* Col-0 genotype which should germinate early in the spring prior to at least decaying fruit. And decaying fruit and animal droppings have nowhere near the ubiquity as light availability or temperature across an ecosystem. The arguments simply feel that they are entirely built around the saying that the model must be true rather than commenting about what information and experiments are needed to really know if the model might be true.

*Reviewer #2:*

The quality of the revised manuscript is improved. I have only one comment on this.

The author argued the genetic and biochemical links between AMB and DELLA. The reviewers request to address the directness of this interaction, which is important to conclude the mode of AMB action. To demonstrate the direct interaction between AMB and DELLA, in vitro experiments such as pull-down or the yeast system is required. The present data set don't answer if AMB directly binds DELLA to block its function. Whatever direct or indirect, this experiment will enhance the value of this work.

Reviewer #3:

The revised version of this manuscript is greatly improved. Notably, the newly added experimental data rule out the hypothesis that AMB prevents seed germination by interfering with ethylene biosynthesis via limitation of methionine levels.

Most of my concerns have been adequately addressed in the authors' pointwise replies and in the edition of the main text. It is notably very relevant to see in Figure 2A that rgl2 mutant seeds exhibit a slight insensitivity to PAO1. Additional data on GAI and RGA are important; notably the fact that AMB does not interfere with the GA-dependent degradation of GAI and RGA, along with RGL2. It remains puzzling how AMB promotes DELLA activity but this could be further dissected in a follow-up study. The strength of this study is indeed that it pinpoints on a novel interaction between AMB-based quorum sensing and seed germination.

Also, in agreement with concerns originally raised by reviewer 1, I recommend authors to avoid, if possible, putting too much extrapolations on the adaptative value of this response. The protective function of this response can only be tested in real field settings and a different outcome may be observed compared to these on-plate germination assays. A recommendation, rather than discussing too much on the protective value of the germination inhibition by AMB, is to more simply comment in the last part of the discussion on the experiments needed to test this hypothesis.

---

## [Author Response]

Summary:This manuscript works to identify a bacterial metabolite that inhibits Arabidopsis germination and proceeds to begin identifying potential mechanisms.Essential revisions:1) Additional evidence on the directness of the AMB/GA link is still unproven considering the previously published AMB/AVG links to ethylene and auxin. This could be provided by either evidence of the directness of the relationship, i.e. does AMB inhibit GA-Dependent Della degradation. Or alternatively assessment of the roles of ethylene or auxin in linking AMB to GA signalling.

In our view the link between AMB and these hormones (GA, ethylene or auxin) can be genetic or biochemical. Therefore, we will discuss here what can be said about these links while presenting the results of additional data in the revised manuscript that speak against an involvement of ethylene or auxin in AMB-dependent germination arrest responses.

1) Links between AMB and GA.

We showed that there is a genetic link between AMB and DELLA factors because mutant seeds lacking functional *DELLA* genes are better able to germinate in presence of AMB (Figure 4C and 4D). Therefore, to the extent that DELLA factors have been related to the phytohormone GA, then there is a genetic link between AMB and GA.

Concerning a potential biochemical link between AMB and the phytohormone GA, we provided two independent lines of evidence that AMB does not interfere with GA synthesis: (1) GA treatment does not trigger seed germination of seeds treated with AMB, suggesting that germination arrest in presence of AMB is not due to limiting seed endogenous GA synthesis (Figure 3A) and (2) the transcriptome of PAC-treated seeds is highly different from that of seeds treated with AMB-containing *P. aeruginosa* extracts (Figure 3C and D, Supplementary file 1 and Supplementary file 2). Therefore, based on these data, there is no evidence that AMB biochemically prevents GA synthesis.

Concerning the hypothesis that AMB interferes with GA-dependent DELLA degradation, we already investigated this matter in the manuscript (mainly in the original Figure 5). We believe that these results, presented in the original Figure 5 were confusing because they mixed two different issues: (1) that AMB mimics the effects observed with *P. aeruginosa* and (2) that AMB is not interfering with GA-dependent DELLA degradation but rather stimulating DELLA activity to promote ABA-dependent responses.

We have now separated these issues in dedicated figures and rewritten this section to hopefully make it clear (changes made to the text are highlighted in yellow in a separate file). We now also included GAI and RGA in the analysis of the effects of AMB.

New Figure 5 shows that AMB mimics the effects observed with *P. aeruginosa*.

New Figure 6 is focused on the mode of action of AMB; it shows that: (1) AMB and PAC do not induce the same effect on RGL2, GAI and RGA protein levels (Figures 6A, 6B, 6C), (2) GA can induce the downregulation of RGL2, GAI and RGA protein levels in presence or absence of AMB (Figure 6D and Figure 6—figure supplement 1) (3) AMB activates ABA-responsive gene expression through DELLA factors (Figure 6E).

The results obtained with GAI and RGA are similar to those observed with RGL2. In both cases our data do not support the hypothesis that AMB prevents GA-dependent RGL2, GAI or RGA degradation.

In a nutshell, the arguments against the hypothesis that AMB interferes with GA-dependent DELLA degradation are:

a) Upon transfer to a PAC-containing medium, the accumulation of DELLA factors RGL2, GAI and RGA markedly increases within hours, consistent with the notion that low GA stabilizes DELLA factors. In contrast, no such marked increase is observed with AMB, indicating that AMB does not interfere with GA-dependent DELLA degradation (new results presented in Figures 6A, 6B, 6C).

b) Similarly, when PAC-treated seeds, accumulating high levels of RGL2, GAI and RGA, are transferred to GA, the levels of RGL2, GAI and RGA similarly decrease irrespective of the presence or absence of AMB. This suggests that AMB does not interfere with GA-dependent DELLA degradation (Figure 6D and Figure 6—figure supplement 1).

We have also rewritten the text describing the results strongly suggesting that AMB stimulates DELLA activity promoting ABA-dependent germination arrest responses (subsection “AMB promotes DELLA factor activity to stimulate ABA signaling”).

2) Links between AMB and Ethylene or Auxin.

As stated in subsection “Oxyviniglycines differentially affect seed germination”, previous reports have biochemically linked AMB to ethylene and auxin: AMB was shown to lower ethylene levels in apples and to inhibit tryptophan synthase in vitro. The effect of AMB in ethylene or auxin synthesis in *Arabidopsis* is not known.

We understand the reviewers’ question as follows: can the DELLA-dependent germination arrest triggered by AMB be the result of a potential effect of AMB on lowering ethylene or auxin levels in seeds? We explicitly tested this hypothesis possibility in the revised version of the manuscript by providing additional experiments. We also addressed the possibility raised by reviewer 2 that AMB interferes with methionine biosynthesis.

In a nutshell, we do not find solid evidence for these hypotheses:

a) ETHYLENE (subsection “Oxyviniglycines differentially affect seed germination”).

Wilson et al. previously reported that 5 µM AVG, a widely used inhibitor of ACC synthase, lowers ethylene synthesis in *Arabidopsis* seeds and does not inhibit germination (Wilson et al., 2014). Here we report that as much as 200 µM AVG did not noticeably inhibit seed germination, further confirming the results of Wilson et al. (Figure 4—figure supplement 10). Furthermore, heptuple *acs* mutant seeds, deficient in *ACS* biosynthetic genes, germinated similarly to WT seeds(Figure 7—figure supplement 4A and Figure 7—figure supplement 4B). Seeds deficient in *ACS* biosynthetic genes would be expected to respond more strongly to AMB-containing WT *PAO*1 extracts if AMB acts by inhibiting ethylene biosynthesis. However, this is not what we observed (Figure 7—figure supplement 4A and Figure 7—figure supplement 4B).

We also treated seeds with silver nitrate (AgNO3), which induces ethylene insensitivity, and did not observe an inhibition of seed germination (Rodriguez et al., 1999) (Figure 7—figure supplement 4C). Altogether, these experiments show that inhibition of ethylene biosynthesis upon seed imbibition is not sufficient to block germination and therefore are not consistent with the hypothesis that AMB blocks germination because it blocks ethylene biosynthesis.

b) AUXIN (see subsection “Oxyviniglycines differentially affect seed germination).

AMB could block germination as a result of its inhibition of auxin synthesis. This possibility is unlikely because (1) auxin promotes ABA-dependent repression of seed germination and (2) low auxin levels may facilitate seed germination since ABA-dependent inhibition of radicle elongations involves enhancement of auxin signaling in the radicle elongation zone (Belin et al., 2009; Liu et al., 2013). We found that *tir1* mutants, deficient in the auxin receptor TIR1, germinate normally, consistent with the report of Liu et al. showing that *tir1/afb2* and *tir1/afb3* double mutants, deficient in the auxin receptors TIR1 and AFB1 or TIR1 and AFB3, germinate normally and are less dormant (Figure 7—figure supplement 5A and B) (Liu et al., 2013). Furthermore, WT seeds treated with 500 µM of yucasin, a potent inhibitor of the YUCCA proteins, which are flavin mono-oxygenases oxidizing indole-3–pyruvic acid to indole-3–acetic acid (auxin), did not prevent their germination (Figure 7—figure supplement 5C) (Nishimura et al., 2014). These data indicate that low auxin signaling or synthesis upon seed imbibition does not prevent seed germination. In addition, *tir1* mutant seed germination responses to AMB-containing WT *PAO1* extracts was similar to that of WT seeds, indicating that AMB-dependent responses in seeds do not require auxin signaling (Figure 7—figure supplement 5A and B).

c) METHIONINE (see subsection “Oxyviniglycines differentially affect seed germination).

AMB could block germination as a result of its inhibition of methionine synthesis. Previous reports showed that methionine biosynthesis is essential for seed germination (Gallardo et al., 2002). We explored whether AMB could block germination by inhibiting methionine synthesis. DL-Propargylgylcine (PAG) is an active site-directed inhibitor of cystathionine γ-synthase, which is necessary for methionine synthesis (Thompson et al., 1982). WT seeds treated with 1 mM PAG were unable to germinate, consistent with previous reports (Gallardo et al., 2002). As expected and consistent with previous reports, exogenously added methionine in the germination medium fully restored germination in PAG-treated seeds (Figure —figure supplement 6) (Gallardo et a.l, 2002). However, exogenous methionine did not rescue the germination of *PAO1* treated WT seeds. These observations are not consistent with the hypothesis that AMB prevents germination by limiting methionine synthesis.

2) How does this mechanism relate to other oxyvinylglycine metabolites that are widespread? Are other AMB/AVG's functioning by a similar pathway or alternative is AMB altering seedling germination in a range of plants where it is in a known pathogen? The limit on one bacteria to one plant inhibits the interpretation of the ecology and evolution.

Oxyvinylglycines are widespread but their biological function is poorly understood.

3) Clean up the citations on cloning the AMB pathway.

OK. See also our detailed response to reviewer 3.

4) Clarify the potential for alternative ecological and evolutionary models that are equally potential.

Concerning potential for alternative ecological and evolutionary models please see our more detailed response to reviewer 2. These alternative models are now presented in the revised version of the manuscript (subsection “Potential ecological and evolutionary significance of an AMB-dependent germination arrest).

Reviewer #1:In this manuscript, the authors describe the identification of an anti-germination metabolite from P. aeruginosa. The metabolite, AMB, affects germination via the Della system. The link of compound to germination appears fairly tight but there are alternative hypothesis that could equally explain the potential ecology/evolution of this connection that are not discussed by the authors. Further, it is not clear what this system may mean in the natural environment. There are additional datastreams that the authors could utilize to support their arguments. Presently, the ecology and evolutionary arguments are not supported.

We thank you for offering alternative hypotheses. We added them in the Discussion section of the manuscript. Our initial proposition that the germination arrest in response to biotic factors serves a protective role echoes the similar and well-accepted notion in the field of seed biology that germination arrest in responses to abiotic factors is a protective response (Kami et al., 2010; Penfield and King, 2009). The germination arrest is interpreted as protective either because it ensures that germination takes care in the proper season (e.g. the case of seed dormancy or seed thermoinhibition) or because it directly protects the seedling (e.g. germination under far red light is blocked presumably because far red light is not favorable for photosynthesis and therefore not favorable for seedling survival).

Seed thermoinhibition is an interesting example: WT *Arabidopsis* seeds imbibed at 34˚C will not germinate (Toh et al., 2008 and references therein). This germination arrest is viewed as “crucial for *Arabidopsis thaliana* to establish vegetative and reproductive growth in appropriate seasons” (Toh et al., 2008). In the laboratory, lowering the temperature will trigger germination, which will yield normal seedlings. In contrast, *aba* mutants, unable to synthesize ABA, or ABA signaling mutants, are unable to control their germination at 34˚C and therefore germinate and die at a percentage of 100% (Lee and Lopez-Molina, unpublished observations). Thermoinhibition is therefore clearly protecting *Arabidopsis* by preventing the formation of the seedling, which would otherwise not survive at 34˚C.

In the same manner, it is tempting to expand the notion of a protective germination arrest to biotic factors, which is what we propose in this manuscript. However, as you rightly point out, this is highly speculative at the present time since to our best knowledge this is the first report showing that a biotic compound is capable of activating the *Arabidopsis* signaling pathways known to control seed germination. More research is needed to further assess the biological, ecological and evolutionary value of the concept of a protective control of seed germination in response to biotic factors.

A central concern that I have about the manuscript is the assumption that AMB inhibiting germination is a "protective" event.

We stand behind our in vitro experiments showing that *Arabidopsis* seeds that do not germinate in presence in presence of AMB are protected, unlike those that germinate (Figure 7). On the other hand, the proposition that plants would have evolved the capacity to respond to AMB in order to protect themselves is a speculation and should not be viewed as a main “result” of this paper. As stated above, the new version of the manuscript includes alternative interpretations. See our comments below.

This arises from several issues. First, does Arabidopsis really follow a fragile seedling ecology? A single plant will produce thousands of seeds and only a single seed needs to proceed to maturity to maintain the population size. So, it is not really clear that the "mammalian" protects your progeny eco-evo thoughts really apply to this situation. This eco-evo thought really only applies when maternal reproduction is limiting on an organism's population growth potential. The authors should better discuss the ecology and evolution of seedling defense under a more accurate representation of the system.

In the case of *Arabidopsis*, we do not think that its high seed numbers excludes protective germination arrest responses to enhance plant survival. Indeed, *Arabidopsis* does not produce highly dispersible seeds (such as dandelion) and most of them are expected to fall in the vicinity of the mother plant. *Arabidopsis* seedlings are very small and fragile. This would increase their chance of being killed at the same time whenever faced by a local threat. There is clear evidence that early development is tightly regulated in *Arabidopsis* to enhance plant survival. Thus, *Arabidopsis* has evolved elaborate germination arrest control mechanisms that are widely regarded as being protective. These include seed dormancy, believed to prevent germination out of season, and control of seed germination of non-dormant seeds in response to abiotic factors, which are also believed to protect the plant (Sano et al., 2016). For example, *Arabidopsis* seedlings can be killed very rapidly by osmotic stress (Lopez-Molina et al., 2001) or by high temperature (see above).

Beyond germination, *Arabidopsis* embryos have evolved the capacity to elongate dramatically in the dark (skotomorphogenesis). One could argue that it is not necessary to evolve this form of development in a plant species producing high seed numbers. Yet, it is difficult to challenge the view that skotomorphogenesis provides an advantage to seeds buried deep in the soil.

We therefore think that high seed numbers do not necessarily exclude evolving protective germination arrest responses to enhance *Arabidopsis* survival.

In the revised version of the manuscript we show that AMB-containing *P. aeruginosa* extracts are able to repress germination and compromise early seedling development of other brassicacea species (*Brassica napus, Lepidium sativum, Sinapsis alba, Brassica rapa, Brassica oleracea, Capsella rubella and Capsella orientalis*), unlike extracts not containing AMB. This suggests that the effect AMB on seed germination is not confined to *Arabidopsis*.

We have now made this point in the Discussion section and we thank you for your suggestion.

We agree that one can find arguments against the notion of a biotic control of seed germination. But one can also find arguments supporting this notion as we do in the manuscript. We certainly do not pretend to decide which are the best arguments for or against the notion of biotic control of seed germination. We agree with you that the best approach is to offer both types of arguments in the Discussion section, which is the case now in the new version of the manuscript.

Secondly, how frequent does P. aeruginosa build up in the environment to the level needed to produce nearly 100 μm concentration surrounding an Arabidopsis seed? Especially as this is controlled by quorum sensing.

Please note that we observed substantial inhibition of germination in presence of 16 µM AMB (Figure 4E). Indeed, the observed GRA of AMB varies depending on the presence or absence of other bacterial secreted molecules (discussed in subsection “L-2-amino-4-methoxy-trans-3-butenoic acid (AMB) is the main GRA 208 released by *P. aeruginosa.”*). In nature, it is likely that AMB will be present with other compounds released by bacteria. To address your question, we performed a rough estimation of the concentration of AMB present in germination plates containing bacteria. We compared the MS profile of liquid extracted from germination plates (10 ml of germination medium) where we streaked WT and ∆*ambE P. aeruginosa* cells as in Figure 1A. As a control we used germination plates containing 100 µM synthetic AMB (see figure below). We found that the peak intensity in the liquid obtained from WT plates was 7-8 fold lower than that from synthetic AMB plates. This suggests that the average concentration of natural AMB secreted by WT *P. aeruginosa* in the germination plate is in the order of μM (µM) concentrations. It is therefore plausible that high AMB concentrations (>10 µM) are present in the direct vicinity of bacteria, which could reflect natural conditions.

**Author response image 1. respfig1:** Identification of AMB in solid plate containing *P. aeruginosa PAO1* by LC-MS/MS analysi*s*. A) Peak height of AMB isolated from different solid plates as indicated in legends. Each dot represents a biological replicate. The different labels above represent the exact value for each replicate. B) Representative peak intensity of AMB feature (*m/z 327.12*) in a representative replicate. Note that AMB signal in plate containing *PAO1* growing bacteria is ≈7-8 fold lower compared to the control plate containing 100 μM AMB.

The authors could survey published microbiome work to assess how often P. aeruginosa is seen at high levels in the rhizosphere. The simple presence of an observed phenomena does not prove that there are hidden massive populations of this bacteria, especially as there are alternative explanations for the data as described below.

The number of *P. aeruginosa* cells building up in the environment is subject to controversy. Thus, Green et al., were able to detect *P. aeruginosa* in 24% of soil samples studied and reported that it multiplied in lettuce and bean under conditions of high temperature and high humidity (Green et al., 1974). They concluded “The results suggest that soil is a reservoir for *P. aeruginosa* and that the bacterium has the capacity to colonize plants during favorable conditions of temperature and moisture”.

On the other hand, Deredjian et al., reported that *P. aeruginosa* has low occurrence in French agricultural soils (Deredjian et al., 2014). However, they were able to detect them in high amounts in various manures, consistent with previous reports.

We think it is reasonable to assume that *P. aeruginosa* will be found in high densities in the rhizosphere particularly if food is available together with the proper moisture or temperature conditions. This could include places where fruits are decaying or where animal droppings are present.

In any case, one should keep in mind that it is not obligatory that control of seed germination would necessarily only take place in presence of high densities of bacteria as observed under laboratory conditions. Indeed, individual bacterial cells release diffusible autoinducers that are perceived by other bacteria. The Quorum Sensing is invoked when the autoinducers released by high densities of cells trigger coordinated responses in cells that would benefit a large group of bacteria.

However, an individual cell is only able to detect the autoinducer concentration present in its immediate proximity, which can include the autoinducer molecules that the same individual cell releases. In the rhizosphere, it is easy to imagine situations limiting autoinducer diffusion, altering autoinducer advection, reducing autoinducer degradation or altering autoinducer spatial distribution, which could lead to high local concentrations of autoinducer even in absence of high densities of bacteria. A given bacteria that only relies on sensing autoinducers in its immediate environment cannot distinguish among the various scenarios leading to high autoinducer concentration. These types of considerations lead Redfield to propose that autoinducers fulfill a role beyond that of detecting high densities of bacteria (Redfield, 2002). They could allow bacteria to sense whether diffusion of molecules in their immediate environment is limited (Diffusion Sensing –DS-). In turn, this would allow a given bacteria to determine whether a given effector would diffuse efficiently or not. A more elaborated discussion of these issues can be found in Hense et al., 2007.

These considerations therefore render plausible that QS responses and indeed high local concentrations of AMB are theoretically possible in the rhizosphere without necessary invoking high bacterial cell densities. In this context, it is not entirely implausible that it could be advantageous for a small seed such as that of *Arabidopsis* to mount a protective germination arrest.

These points are now included in the Discussion section of the revised manuscript.

Part of the solution to this concern is that AMB is a part of a larger family of oxyvinyl glycines. These compounds are known from a number of plant associated bacteria and they are thought to inhibit methionine metabolism. This raises two issues, the first is that the authors could support their argument about protective effect by doing a broader survey of plant associated bacteria for oxyvinyl glycine production and test one or two additional mimetics. This would allow them to expand their arguments vastly beyond P. aeruginosa and make a more ecological argument. Further, it would mean that it is the level of total oxyvinyl glycines being produced in an environment and not those solely from P. aeruginosa.However, equally, the fact that these compounds are likely methionine synthesis inhibitors. This will conversely affect ethylene production which agrees with the pictures in Figure 6A with altered apical hook formation. This means that the authors need to consider the potential that the AMB is really stimulating the known Ethylene to Della links via an alteration in plant amino acid metabolism.

In the Discussion section we mention the various cases where a biological or biochemical activity has been associated with oxyvinylglycines produced by different bacteria. We also indeed mention that they could act as methionine antimetabolites and therefore function as methionine synthesis inhibitors. However, the biological function of oxyvinylglycines remains largely unknown.

In the Results section we had already described the effect of the oxyvinylglycine AVG but we omitted to mention that it was a well-known inhibitor of ethylene synthesis (we only mentioned it in the Discussion section). In the revised manuscript we now also mention in the Results section that AVG is an inhibitor of ethylene synthesis. We reported that as much as 200 µM of AVG does not inhibit *Arabidopsis* seed germination. This is consistent with previous publications. For example in Wilson et al. (Wilson et al., 2014) it is mentioned “We found that application of AVG lowered ethylene production (data not shown); however, this had no measureable effects on the time required for 50% of Col or etr1-6 seeds to germinate (Figure 6A), which is similar to what has been observed previously for several other species (Rudnicki et al., 1978; Fu and Yang, 1983; Hoffman et al., 1983; Kepczynski and Karssen, 1985; Abeles, 1986; Iglesias-Fernández and Matilla, 2010)”.

Other researchers have indirectly associated the oxyvinylglycine 4-formylaminooxyvinylglycine (FVG), produced by *Pseudomonas* fluorescens strain WH6, with a germination repressive activity but the concentrations used were not specified (this is already discussed in the text). We could not obtain the pure synthetic compound to test its germination repressive activity in *Arabidopsis*. Importantly, FVG is reported to not be active in dicots to repress germination (Banowetz et al., 2008). However, we reported in the revised version the effect of AMB-containing extracts on several brassicaceae seeds. These new data suggest that the oxyvinylglycines AMB and FVG affect differently germination of a given species (such as cabbage, which is strongly repressed by AMB containing extract but only mildly by FVG) (Banowetz et al., 2008).

Previous reports showed that methionine biosynthesis is essential for seed germination (Gallardo et al., 2002). To further assess the hypothesis that AMB blocks germination by interfering with methionine metabolism, we now have included in the revised manuscript the use of DL-Propargylgylcine (PAG), an inhibitor of methionine biosynthesis. PAG is not an oxyvinylglycine but is an active site-directed inhibitor of cystathionine γ-synthase catalyzing the sulfur-linked joining of cysteine and homoserine 4-phosphate to form cystathionine, which is needed to form homocysteine. In turn, methionine synthase converts homocysteine to methionine (Ravanel et al., 1998).

We found that 1 mM PAG indeed inhibits *Arabidopsis* germination consistent with previous reports (Gallardo et al., 2002). As expected, exogenously added methionine in the germination medium fully restored germination in PAG-treated seeds (Figure 7—figure supplement 6). We found that exogenous methionine did not restore the germination of AMB-treated WT seeds. This is not consistent with the hypothesis that AMB prevents germination by limiting methionine synthesis.

Altogether, the data with AVG and PAG are now included in the revised manuscript are not supporting the view that AMB exerts its DELLA-dependent seed germination arrest by inhibiting methionine or ethylene synthesis.

The presented data shows that AMB works via Della but not that it interacts with Della.

We never claimed that AMB interacted with DELLA factors. We only proposed that AMB stimulates in an unknown manner DELLA activity to stimulate endogenous ABA signaling. This DELLA activity was only characterized genetically (Piskurewicz et al., 2008) and its nature is not known.

This is key to the story as if this is the mechanism, then it changes the interpretation from the Plant evolving a connection to the bacteria simply tapping into a mechanism that already existed in the plant.

We agree with you that we cannot exclude that AMB happens to be a molecule interfering with an (unknown) mechanism already present in the plant that is linking amino acid metabolism with DELLA factors or any other GA-independent mechanism involving DELLA factors to control germination. In this alternative view there is no need to invoke evolutionary mechanisms linking biotic factors with control of seed germination.

This alternative view is now included in the Discussion section of the revised manuscript.

The authors should comment on if their Psedomonad species individuals in Figure 1 have an oxyvinyl glycine operon. Optimally, they would query across the available genomes to see if other members of these species have an operon.

As mentioned in the Discussion section, the biological function of oxyvinylglycines is not well understood. Furthermore, the synthesis of different oxyvinylglycines can follow completely different enzymatic steps. This is easily seeing by comparing the different gene products responsible for FVG, AVG, rhizobitoxine or AMB synthesis so it is extremely difficult to scan genomes for operons generally involved in oxyvinylglycine synthesis.

Concerning the *amb* operon, which is genetically necessary for AMB synthesis and release, we already mentioned in the text that the *amb* operon is only present in *Pseudomonas aeruginosa.* Note that there is a controversy concerning this operon as one group proposed that *amb* gene products are involved in the synthesis and release of AMB whereas another group proposed that they are involved in the synthesis of IQS (this is discussed in the text).

The *gvgRABCDEFGHI* operon responsible for FVG production is present in *P. fluorescens* strain WH6 (Okrent et al., 2017). Interestingly, Trippe and co-workers found that the operon was present in several *P. fluorescens* strains, but not all. For example, it is not present in *P. fluorescens* strain CHA0 (Kristin M. Trippe, personal communications). They also found that the *gvgRABCDEFGHI* operon is present in *P. syringae* where it is not active as FVG could not be detected (Kristin M. Trippe, personal communication). The *gvgRABCDEFGHI* is also absent in *P. aeruginosa*.

The *rtxA* and *rtxC* genes are necessary for the production of the oxyvinylglycines rhizobitoxine in *Pseudomonas andropogonis* (Yasuta et al., 2001).

To our best knowledge, the genes coding for the proteins directly responsible for the synthesis of AVG have not been described. However, mutants of the *Streptomyces* strain NRRL 5331 that are unable to synthesize AVG have been described (Cuadrado et al., 2004; Fernández et al., 2004).

Concerning other operons involved in the production of oxyvinylglycines in other *Pseudomonas* species, we could not find any information in the website www.*Pseudomonas*.com.

RNAseq – It is not clear if there is any replication in the RNAseq analysis. The level of independent biological replication needs to be provided to the reader to allow them to evaluate the data. The same is equally true of the RT-qPCR analysis as it is not clear what is an independent biological versus technical replicate. The two need to be handled very differently.

RNAseq analysis was performed with two independent biological replicates (indicated in MMaterials and methods section and figure legend). The RT-qPCR results were performed with three independent biological replicates (indicated in figure legend).

Subsection “How AMB interferes with GA signaling remains to be understood” – Please remember that Arabidopsis does not have fire ecology and no need for smoke compounds to affect germination. As such, the Karrikin to seed relationship in Arabidopsis has no link to smoke.

We did not intend to say that *Arabidopsis* has a fire ecology. However, karrikins were reported to promote *Arabidopsis* seed germination and in this context researchers identified the protein KAI2 which binds karrikins and is commonly referred in the literature as a “karrikin receptor” that is specifically required for *Arabidopsis* responses to karrikins (Waters et al., 2012; Guo et al., 2013). In our RNAseq analysis we identified *Arabidopsis* genes annotated as involved in “karrikin signaling” and their products could be targets of AMB.

Reviewer #2:Chahtane et al., report that AMB secreted by P. aeruginosa blocks DELLA-dependent GA signaling, which in turn inhibits Arabidopsis germination. The authors show AMB synthesis in P. aeruginosa is controlled by the quorum sensing system. The authors also show that the germination arrest is a protective mechanism for plants to avoid severe seedling damage.This work investigated seed germination under biotic stress conditions. They tested the germinability of Arabidopsis together with four Psuedomonas species and E. coli and found only P. aeruginosa grown close to seeds inhibit seed germination.Both bacterial genetic approach and metabolomic approach identified AMB, an oxyvinylglycine, is the primary chemical to inhibits seed germination. It has been shown that AMB released by P. aeruginosa inhibits growth of Erwinia, and some oxyvinylglycines inhibit seed germination.This work includes interesting findings. Oxyvinylglycines released by Psuedomonas have been shown to alter the growth of neighbors, including seeds. It seems that Psuedomonas utilizes AMB to control other organisms surrounding itself. These previous publications provide a credibility of this work to be biological relevant.The authors discuss AMB acts through the DELLA-dependent pathway (and also GA-independent other pathways). This work will be more attractive if the authors investigates how AMB inhibits GA-mediated DELLA accumulation. The biochemical analysis to investigate if DELLA accumulation observed is reduced degradation or enhanced protein production would be worth to be included, even though the authors are not able to specify the target(s) of AMB.

See our detailed response to the editor and reviewer 3. We have provided experiments (notably in Figure 6D and Figure 6—figure supplement 1) suggesting that AMB does not affect the GA-dependent DELLA protein degradation. The text describing Figure 6 has been improved. In the revised version of the manuscript we have included RGA and GAI in the analysis. Our results are not supporting the notion that AMB interferes with GA-dependent DELLA degradation. Rather they suggest that AMB stimulates DELLA activity.

Reviewer #3:This study investigates one mechanism by which Arabidopsis seeds eavesdrop on the presence of bacterial pathogens in the soil. Two levels of analyses are considered: (i) the chemical nature of the germination-inhibiting small molecule (AMB) (as well as the dependency of AMB synthesis on the 5-gene operon ambABCDE) produced by Pseudomonas aeruginosa (PAO), a plant pathogen whose presence in close proximity to Arabidopsis seeds blocks their germination; and (ii) the mechanism by which AMB modulates GA signaling and thereby germination. Specifically, authors provide clear evidence that AMB is the central small molecule responsible for the strong germination repressive activity (GRA) of PAO on Arabidopsis seeds. Furthermore, using Western blot analyses for RGL2 (a DELLA factor with central function during germination) and ABI5 (an RGL2-controlled ABA signaling transcription factor known to repress germination) as well as germination bioassays with related mutants, authors demonstrate that the GRA of PAO is linked to a prolonged RGL2 stability that promotes the accumulation of ABI5.

We did not phrase it like this, please see below.

Overall the study is an interesting merger between germination bioassays, the use of signaling mutants and metabolomics of bacterial natural products to address seeds' germination arrestment in response to perception of soluble bacterial cues. Nonetheless, a weakness of this study is that it does not provide a clear mechanism for how AMB interferes with GA signaling and that it is completely based on assumption that "RGL2 can be used a representative DELLA factor" in the context of this germination arrestment response, while germination of rgl2 as shown in Figure 2A is still strongly compromised by the presence of PAO.The supplemental material dealing with the differential metabolomics studies of bacteria-free extracts, isolation of the GRA-causing molecules, chemical identification and molecular networking analysis of AMB clearly supports the identification of AMB as the key metabolite responsible for the GRA of PAO on Arabidopsis seeds. However, while interesting in order to establish links between seed germination arrestment and bacterial quorum sensing, the part dealing with the dependency of AMB synthesis on the 5-gene operon ambABCDE is essentially confirmatory to previous published work referenced in the text as Lee et al., 2010b and 2013b. In the current version of the text, this did not readily appear clear to me. At least, authors should more rigorously report on previous studies dealing with the operon-dependent synthesis of AMB.

We fully agree with you that our finding that the 5-gene operon *ambABCDE* is necessary for AMB synthesis confirms previous published work. We did not intend to claim that this is a finding of our work. Indeed, in the first version of the manuscript we wrote (subsection “AMB production and release is under the control of *P. aeruginosa*’s quorum sensing IQS”) “there is no genetic controversy regarding the need of a functional ambABCDE operon for (1) IQS production and signaling and (2) AMB synthesis and release by *P. aeruginosa*. In this study, we further confirm that presence of the ambABCDE operon is necessary for AMB production”.

In the new version of the manuscript we make this point even clearer by including the Lee et al., 2010b and 2013b references in the as follows: “AMB production and release is dependent on the five-gene operon ambABCDE controlling the newly identified quorum-sensing IQS in *P. aeruginosa* (Lee et al., 2010, 2013a).” and in the DDiscussion section as follows: “…we further confirm that presence of the ambABCDE operon is necessary for AMB production (Figure 4—figure supplement 5, Table S3)” and “Indeed Lee et al.. showed that ambABCDE is also necessary for AMB production and Rojas Murcia et al. proposed that ambABCDE gene products rather synthesize and export AMB (Rojas Murcia et al., 2015; reviewed in Moradali et al., 2017; Lee et al., 2010b, 2013a)”.

On the other hand, the finding that the *las* and *rhl* QS subsystems are required for AMB synthesis is, to our best knowledge, novel (line 547-549; Figure 4—figure supplement 5A, Table S3).

A striking result of this study is that the GRA of PAO is almost fully abolished for mutant seeds lacking all five DELLA factors. This is a key result as it establishes a link between the GRA of PAO (and AMB) and GA signaling. However, in the same figure panel, it is also clear that germination success only mildly increased (compared to that of WT seeds) for rgl2 seeds growing in proximity of PAO. Therefore, regulation of RGL2 levels is not sufficient to explain germination arrestment by PAO. According to the mechanism proposed by this study to explain the GRA of PAO, that AMB-triggered prolonged RGL2 stability promotes the accumulation of the repressing factor ABI5, one would have expected a much stronger germination rate for PAO-challenged rgl2 seeds. In my view, this challenges the use of RGL2 as unique representative DELLA factor throughout the study. Hence, it would have been interesting to monitor, in the context of the different germination/treatment bioassays, levels of other DELLA proteins.

We did not claim that regulation of RGL2 levels was sufficient to arrest germination in response to *PAO1*. Nor did we propose that the prolonged RGL2 accumulation in seeds exposed to *PAO*1 was only responsible for the accumulation of ABI5. We propose that AMB promotes “DELLA-dependent increase of the germination repressor ABI5, as previously shown with seeds unable to synthesize GA upon imbibition (Piskurewicz et al., 2008, 2009)” (subsection “The GRA released by *P. aeruginosa* stimulates GA and ABA signaling pathways to repress seed germination” and “…AMB stimulates the DELLA activity promoting ABA signaling in seeds” (subsection 2AMB promotes DELLA factor activity to stimulate ABA signaling).

The importance of RGL2 among other DELLA factors to control seed germination comes from the fact that its mRNA is positively regulated by ABA, which leads to strong RGL2 accumulation relative to other DELLA factors (Piskurewicz et al., 2008). Thus, *ga1/rgl2* seeds or PAC-treated *rgl2* seeds can germinate in white light unlike *ga1/rga, ga1/gai* of *ga1/gai/rga* seeds or PAC-treated WT, *gai* or *rga* seeds where RGL2-dependent accumulation of ABI5 blocks germination (Lee et al., 2002; Piskurewicz et al., 2008).

However, we previously showed that RGL2 is not the only factor promoting ABA synthesis and signaling in seeds. Other DELLA factors, such as RGA and GAI, can do so in an additive manner (Piskurewicz et al., 2008, 2009). Under far red light, WT and *ga1/rgl2* cannot germinate due to ABA accumulation in seeds. This is because under far red light PIF1/PIL5 is stable and can promote *RGA* and *GAI* mRNA expression, which allows for sufficient RGA and GAI accumulation to promote accumulation of the ABA-responses factors ABI3 and ABI5 (*ga1/rgl2/rga/gai* seeds can germinate under far red light due to low ABA-dependent responses) (Piskurewicz et al., 2009).

Based on these reports, there is therefore no need to expect that *rgl2* mutants should be insensitive to AMB or *PAO1* since other DELLA factors can promote ABI5 accumulation (note: In the revised manuscript, Figure 2A includes now data from five biological replicates that seem to indicate a slight insensitivity of *rgl2* mutant seeds to *PAO1*, however they are not statistically significant).

In the first version of the manuscript we used RGL2 as a representative DELLA factor to study how AMB could elicit DELLA-dependent germination arrest. We provide evidence that AMB does not interfere with GA-dependent RGL2 degradation and we extrapolated this notion to other DELLA factors given that the machinery responsible for GA-dependent degradation is the same for all DELLA factors, according to the current model.

To further verify this claim we have now included in the revised version of the manuscript data for GAI and RGA. In a nutshell, as for RGL2, we do not see any evidence that AMB increases the stability of RGA and GAI. We include below our comments to the editor regarding this point:

a) Upon transfer to a PAC-containing medium, the accumulation of DELLA factors RGL2, GAI and RGA markedly increases within hours, consistent with the notion that low GA stabilizes DELLA factors. In contrast, no such marked increase is observed with AMB, indicating that AMB does not interfere with GA-dependent DELLA degradation (new results presented in Figures 6A, 6B and 6C).

b) Similarly, when PAC-treated seeds, accumulating high levels of RGL2, GAI and RGA, are transferred to GA, the levels of RGL2, GAI and RGA similarly decrease irrespective of the presence or absence of AMB. This suggests that AMB does not interfere with GA-dependent DELLA degradation (new results presented in Figure 6D and Figure 6—figure supplement 1).

We rather provide evidence that AMB is stimulating DELLA activity to promote ABA-dependent germination arrest responses (Figure 6E). We have also tried to improve the text describing these results (subsection “AMB promotes DELLA factor activity to stimulate ABA signaling”).

Along with this, are there any other DELLA genes present in the list of 130 PAO1-regulated genes?

We respectfully do not understand what you mean by “any other DELLA genes”. We did not specifically mention the regulation of DELLA genes in the RNAseq experiment. The goal of the RNAseq experiment was to compare very early gene expression changes elicited by PAC and *PAO1* extracts. At this early time points DELLA gene expression is not markedly altered in PAC-treated seeds, except for *GAI* mRNA, which is repressed consistent with previous reports (Piskurewicz et al., 2008). Similarly, there are no significant changes in DELLA gene expression in the list of 130 *PAO1*-regulated genes.

Interestingly, it is known from previous studies that a prominent plant target of AMB is the ethylene biosynthesis enzyme ACC synthase. Ethylene is required in seeds of many species for dormancy release or germination under optimal or adverse conditions. In particular, ethylene is known to inhibit ABA signaling in seeds to allow germination. The hypothesis that a direct inhibitory activity of AMB over ethylene synthesis could be responsible for the deregulation in ABA signaling observed in non-germinating seeds is not discussed in this study. Did the authors identify signatures of ethylene signaling deregulation in the PAO transcriptome data-set?

We did not find a signature for ethylene signaling in the *PAO1* transcriptome data set (Supplementary file 2, see Gene Ontology analysis). In the revised version of the manuscript we address the hypothesis that AMB could block germination by inhibiting ethylene synthesis and our data do not support this hypothesis. We also discuss the hypothesis that AMB could arrest germination by blocking auxin or methionine biosynthesis and our findings do not support this conclusion (see our comments to the editor). Concerning the ethylene synthesis hypothesis, see below an excerpt of our answer to the editor:

Wilson et al., previously reported that 5 µM AVG, a widely used inhibitor of ACC synthase, lowers ethylene synthesis in *Arabidopsis* seeds and does not inhibit germination (Wilson et al., 2014). Here we report that as much as 200 µM AVG did not noticeably inhibit seed germination, further confirming the results of Wilson et al., (Figure 4—figure supplement 10). Furthermore, heptuple *acs* mutant seeds, deficient in *ACS* biosynthetic genes, germinated similarly to WT seeds (Figure 7-figure supplement 4A and Figure 7—figure supplement 4B). Seeds deficient in *ACS* biosynthetic genes would be expected to respond more strongly to AMB-containing WT *PAO1* extracts if AMB acts by inhibiting ethylene biosynthesis. However, this is not what we observed (Figure 7—figure supplement 4A and Figure 7—figure supplement 4B).

We also treated seeds with silver nitrate (AgNO3), which induces ethylene insensitivity, and did not observe an inhibition of seed germination (Rodriguez et al., 1999, Figure 7—figure supplement 4C). Altogether, these experiments show that inhibition of ethylene biosynthesis upon seed imbibition is not sufficient to block germination and therefore are not consistent with the hypothesis that AMB blocks germination because it blocks ethylene biosynthesis.

[Editors' note: further revisions were requested prior to acceptance, as described below.]

Summary:This work begins to study how cross-kingdom signals are sent from bacteria to plants to alter seed germination.Essential revisions:1) There is a technical concern in that the RNAseq is still not fully described.

We provided in the Materials and methods section the additional information requested by reviewer 1 (“Differentially expressed genes were selected according to their significance in fold-change expression (false discovery rate, FDR<0.05) and a threshold level of at least two-fold change between samples (log2 ratio ≥1 and ≤−1”).

2) The writing about DELLA/AMB needs to be clarified at each instance to show that this is solely a genetic link at this point and has not yet been proven to be a mechanistic link.

We modified the manuscript so that at each instance where AMB and DELLA factors or genes are discussed together it is explicitly stated that their link is genetic and not mechanistic. The modifications appear in the Abstract, Introduction and throughout Results section and Discussion section.

3) The ecological and evolutionary ramifications need further qualifications as per the reviewers’ comments.

We respect reviewer 1’s concerns and we also agree with reviewer 3’srecommendations about being cautious regarding the adaptive value of the germination responses reported in the manuscript.

Therefore, in the new revision, we no longer present the concept of “protective germination arrest in response to biotic factors” as the main one we wish to defend.

Rather we discuss this possibility only a possibility equally valid among other possibilities such as the “evolutionary serendipity” mentioned by reviewer 1.

These modifications appear notably in:

1) The Title where we have removed the word “protective”.

2) The Abstract: Only toning down modifications are shown. The Abstract was otherwise slightly modified to avoid to excessive wording.

3) The Introduction: For example, the sentence “Little is known about whether seeds detect biotic factors released by non-plant organisms to control seed germination.” is replaced by “Little is known about whether biotic factors released by non-plant organisms induce seed germination responses in plants.”

The end of the Introduction is also substantially modified.

4) The Discussion section: The last part of the Discussion section dealing with the potential ecological and evolutionary significance of an AMB-dependent germination arrest has also been modified:

a) We start by answering reviewer 1’s question concerning the occurrence of the amb operon in *P. aeruginosa* strains other than the one used in our study. The operon is indeed present in numerous other strains (new Figure 4—figure supplement 11).

b) We continue by discussing the question of the biological significance of the AMB- and DELLA-dependent germination arrest.

Two categories of significance are discussed: (1) the “evolutionary serendipity” case, i.e. a fortuitous effect of AMB on seed germination, which we discuss as still being biologically very interesting because they reveal that AMB interferes with an unknown GA-independent mechanism involving DELLA factors) and (2) the “adaptive case”, i.e. an ecologically and evolutionary significant case.

c) We have taken into account the reviewer 1’s and reviewer 3’s recommendations, including the importance of understanding the cellular ecology of the pathogen in real field settings and the use of mutants in order to test the “adaptive case” model. These appear at the end of the Discussion section.